# EXPLOITING CODE SYMMETRIES FOR LEARNING PROGRAM SEMANTICS

## ABSTRACT

Large Language Models (LLMs) hold significant potential for automating program analysis, but current code LLMs face challenges in grasping program semantics. Our paper addresses this by formalizing program semantics through code symmetries and integrating them into LLM architectures for code analysis. We introduce a group-theoretic framework that defines code symmetries as semantics-preserving transformations, enabling precise reasoning within LLMs. Our solution, SYMC, employs a novel variant of group-equivariant self-attention that is provably equivariant to code symmetries. We extensively evaluate SYMC on four program analysis tasks, comparing it to eight baselines against eight code transformations. Our results show that SYMC generalizes to unseen code transformations, outperforming the state-of-the-art code models by 30.7%. SYMC, by design, stays invariant to semantics-preserving permutations, while code LLMs like WizardCoder and GPT-4 violate these invariances at a high rate (i.e., 14% and 43%, respectively).

## 1 INTRODUCTION

Automated program analysis using Code Large Language Models (LLMs) has become widely popular for software engineering and security tasks (Liu et al., 2023; Maniatis & Tarlow, 2023), but current code LLMs struggle with generalization to new code (Henke et al., 2022; Rabin et al., 2021; Gao et al., 2023b;a; Yefet et al., 2020; Bundt et al., 2022; Zhang et al., 2023). This paper aims to enhance LLMs by establishing and preserving fundamental code symmetries, drawing inspiration from translation and rotation symmetries that typically hold in vision (Cohen & Welling, 2016).

**Code symmetry.** Intuitively, symmetry of code refers to any transformation applied to a code block that preserves the semantics (i.e., input-output behavior) of the original code. Consider a (sequential) code fragment `x=2;y=4`. Reordering the instructions to `y=4;x=2` does not change the semantics of the code. Of course, any code analysis task that depends solely on the semantics of the code (e.g., bug detection) needs to preserve these symmetries by staying invariant to the transformations. Formally, given a code block $c$ and a set of symmetries $G$, an LLM $m$ should ensure $\forall g \in G, m(g(c)) = m(c)$.

**Limitations of existing approaches.** A popular way to train LLMs to be invariant to code symmetries is data augmentation and pre-training (Luo et al., 2023; Feng et al., 2020). However, this approach is not very effective due to the sheer number of possible symmetries and their compositions. Specifically, this approach has two major limitations: (1) it is prohibitively expensive to enumerate each possible variant $g(c)$; and (2) it provides no guarantee of invariance even for the seen symmetries. In fact, we find that existing state-of-the-art LLMs break desired invariances at an alarmingly high rate (e.g., 14% in WizardCoder and 43% in GPT4 as shown in Table 1) even for simple code symmetries like a single statement permutation.

**Our approach.** We introduce a group-theoretic framework to precisely define code symmetries in terms of semantics-preserving permutations of statements and create LLM architectures that inherently preserve these symmetries. Using this framework, we present SYMC, an LLM architecture designed to guarantee invariance to semantics-preserving statement permutations. This is achieved through a $G$-equivariant code representation learning ($r$) followed by a $G$-invariant predictive learning ($p$), with $G$ determined based on the graph automorphisms of the code block's interpretation graph (a generalization of program dependence graph).

Table 1: Invariance violation rate across different code models (darker colors indicate more violations).

| | Violation |
|---|---|
| SYMC | 0% |
| GPT-4 | 43% |
| WizardCoder | 14% |
| code2vec | 61% |
| code2seq | 52% |
| GGNN | 7% |

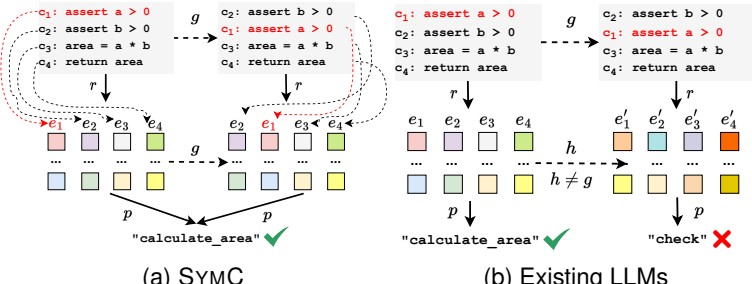

(a) SYMC       (b) Existing LLMs

Figure 1: (a) SYMC as a $G$-invariant function name predictor where $G$ is a group of semantics-preserving statement permutations $g$. (b) Code LLMs not preserving the symmetries in $G$ and thus incorrectly change the output.

Figure 1a shows a concrete example of the benefit SYMC when deployed for function name prediction. The code snippets illustrate a semantics-preserving statement reordering. SYMC enforces its output to stay invariant via keeping its learned representation $G$-equivariant, where the code representation $(e_1, e_2, e_3, e_4)$ is transformed into $(e_2, e_1, e_3, e_4)$, followed by a $G$-invariant prediction module. By contrast, Figure 1b shows an existing code model (Jin et al., 2022) that does not preserve permutation symmetry. In this case, the code representation $(e_1, ..., e_4)$ is transformed into a completely different set of embeddings $(e'_1, ..., e'_4)$, leading to a changed prediction.

**Result summary.** We evaluate SYMC on four program analysis tasks against various source code and binary analysis baselines. For semantics-preserving permutations, SYMC is guaranteed to stay invariant while the state-of-the-art code LLMs like WizardCoder and GPT-4 violate the invariance at a high rate, i.e., 14% and 43%, respectively. For other semantics-preserving source code transformations beyond permutations, SYMC surpasses the state-of-the-art code baselines, namely WizardCoder and GPT-4, by 1.63% and 16.1%, respectively, while maintaining a model size 108× smaller. On sophisticated transformations introduced by compiler optimizations and obfuscations, SYMC outperforms the extensively pre-trained binary analysis baseline, PalmTree, by 30.7% without requiring any pre-training.

**Contributions.** (1) We establish a foundational framework using semantics-preserving code symmetries, ensuring provable generalization to new samples resulting from their compositions. (2) We introduce a mechanism to identify symmetries in programs through graph automorphism. (3) We present a novel LLM architecture with a new variant of self-attention equivariant to program symmetries. (4) Our approach demonstrates effectiveness in generalizing invariance to various semantics-preserving transformations, surpassing state-of-the-art code LLMs in program analysis.

## 2 PRELIMINARIES

This section briefly describes the symmetry groups. See Appendix A for a more formal description.

**Symmetry group.** A *symmetry group* $(G, \circ)$ consists of a non-empty set $G$ of transformations and a binary operator $\circ : G \times G \to G$, where $\circ$ operates on two elements (i.e., transformations) in $G$, e.g., $x, y \in G$, and produces a new transformation $z = x \circ y, z \in G$. The binary operator has to be associative, invertible, and there exists an identity $\exists \mathbf{1} \in G, \forall x \in G, x \circ \mathbf{1} = \mathbf{1} \circ x$.

**Group action.** The elements of a $G$ are abstract transformations that become concrete when they *act* on some set $X$, i.e., they transform $x \in X$ into $x' \in X$ while keeping some properties of $x$ *invariant*. Formally, an action $\bullet$ of a symmetry group $G$ is a binary operation defined on a set of objects $X$, i.e., $\bullet : G \times X \to X$, where it is also associative and has an identity.

It is common in the group theory literature to use $\circ$ to denote both *action* and *composition*, when it is clear from the context (Higgins et al., 2018). It is also customary to interchange $g(x)$ and $g \circ x$. Therefore, we treat $g \bullet (h \bullet x)$, $g \circ (h \circ x)$, and $g(h(x))$ as the same in the rest of this paper.

**Invariance and equivariance.** A symmetry group comes with two properties, namely *invariance* and *equivariance*, that formalize the concept of preservation of some properties when a set $X$ is acted upon by $G$. Let $f$ be a function that maps each element $x \in X$ to a corresponding element $y$ in the set $Y$, indicating the property's value for that particular element. $f$ is called $G$-**invariant** if $\forall g \in G, \forall x \in X, f(g \circ x) = f(x)$. $f$ is called $G$-**equivariant** if $\forall g \in G, \forall x \in X, f(g \circ x) = g \circ f(x)$.

Given the definition of $G$-equivariant function, we have the following lemmas:

**Lemma 1.** Let $f_1$ and $f_2$ be two functions that are both $G$-equivariant and $h = f_1 \circ f_2$ be the new function composed by $f_1$ and $f_2$. $h$ is also $G$-equivariant.

**Lemma 2.** Let $f_1$ and $f_2$ be two functions where $f_1$ is $G$-equivariant and $f_2$ is $G$-invariant, and $h = f_2 \circ f_1$ be the new function composed by applying $f_1$ and then $f_2$. $h$ is $G$-invariant.

## 3 METHODOLOGY

This section describes the definitions of the equivariance/invariance properties for code models, the construction of group-equivariant self-attention layers, and the group-invariant code analysis model.

### 3.1 INVARIANCE & EQUIVARIANCE FOR CODE MODELS

**Code representation units.** We establish formal definitions of the code space as a collection of code blocks, which serves as the input space for representation learning. We then proceed to define representation learning and predictive learning in the code space.

**Definition 3.1.** A **code representation unit** (e.g., procedure) $c$ consists of $n$ instructions from an instruction set $I$, i.e., $c \in I^n$. The **code space** $I^n$ is the set of all code representation units of interest.

A typical Code Representation Unit (CRU) is a method with well-defined interfaces, ensuring controlled interaction with other methods, without arbitrary control transfers. Below, we provide formal definitions for learning program representation and predictive learning.

**Definition 3.2. Representation learning for code** involves learning a function $r$ that maps a CRU $c \in I^n$ to a point in the code representation space $\mathbb{R}^{d \times n}$, $r : I^n \to \mathbb{R}^{d \times n}$, where $\mathbb{R}$ denotes the set of real numbers, and $d$ denotes the dimension of the vector to which each instruction is mapped.

**Definition 3.3. Predictive learning for code** entails learning a function $p : \mathbb{R}^{d \times n} \to \mathbb{R}^L$ that maps the code representation produced by the representation learning $r$ to a label space $\mathbb{R}^L$. $\mathbb{R}^L$ becomes concrete in the context of downstream analyses (§5).

In this framework, the earlier layers of the neural network serve as the representation learning function $r$, learning program representations. The subsequent layers serve as the predictive learning function $p$, making predictions based on analysis-specific labels, such as function names. Therefore, the whole network computation can be thought of as a composition of $r$ and $p$, i.e., $p \circ r$.

In the following, we establish formal properties for code analysis models with explicit representation learning $r$ and predictive learning $p$ based on $G$-equivariance/invariance.

**Definition 3.4** ($G$-**equivariant code representation learning**). Let $G$ be a symmetry group consisting of *semantics-preserving transformations* applied to a CRU $c \in I^n$. A representation function $r : I^n \to \mathbb{R}^{d \times n}$ is $G$-equivariant if for every $g \in G$ and $c \in I^n$, we have $g \circ r(c) = r(g \circ c)$.

Note that here the input space of $r$ ($I^n$) and its output space ($\mathbb{R}^{d \times n}$) are both sets of size $n$, where each instruction $I$ is mapped to a $R^d$ vector by the representation function. This consideration is necessary to ensure the symmetry group can act on both sets appropriately.

**Definition 3.5** ($G$-**invariant code predictive learning**). Let $G$ be a symmetry group consisting of *semantics-preserving transformations* applied to program representation vector $c \in I^n$. A predictive learning function $p : \mathbb{R}^{d \times n} \to \mathbb{R}^L$ is $G$-invariant if $\forall g \in G, \forall e \in \mathbb{R}^{d \times n}, p(g \circ e) = p(e)$.

Stacking $p$ on top of $r$, $p \circ r$, leads to a $G$-invariant model according to Lemma 2.

### 3.2 SEMANTICS-PRESERVING PROGRAM SYMMETRIES

A semantics-preserving program symmetry is a program transformation preserves the input and output behavior of a CRU when interpreted by the program interpretation function $f$. The program

interpretation function takes a CRU $c \in I^n$ as input, where $\mathcal{I}$ represents the set of possible input values to execute CRU, and produces output values represented by the set $\mathcal{O}$.

**Definition 3.6.** A **semantics-preserving program symmetry** $g$ is a transformation acting on $c \in I^n$ ($g : I^n \to I^n$) such that $\forall in \in \mathcal{I}, \forall out \in \mathcal{O}, f(g \circ c, in) = f(c, in) = out$.

**Definition 3.7.** A semantics-preserving **program symmetry group** $G$ is a set of semantics-preserving program symmetries that also satisfy the group axioms.

**Local and global program symmetry.** We call $g$ *local program symmetry* because it acts on a single CRU $c \in I^n$. In this paper, we do not consider *global* program symmetry defined over the entire program space $I^n$, e.g., rotation as in image space. This is because each independent sample $c$ will have its own symmetries corresponding to its semantics-preserving transformations, and we develop model architectures that preserve the specific code symmetry group for each individual sample (§4).

## 3.3 $Aut(\mathcal{IG})$: A Program Symmetry Group

In this paper, we focus on a specific symmetry group that maintains the structural integrity of CRUs by utilizing their inherent compositional structure. However, note that this approach is not the only way to form code symmetry groups and does not encompass all possible code symmetries. We leave further exploration in these directions to future research.

Next, we describe the compositional structure of the program interpreter $f$ operating on a CRU, enabling us to define the program interpretation graph that links CRUs to their input-output behavior.

**Compositional structure of program interpreter $f$.** The interpreter function $f$ (defined in §3.2) can be represented as a composition of individual per-instruction interpreter functions $\{f_1, ..., f_n\}$. Each $f_i : \mathcal{I}_i \to \mathcal{O}_i$ interprets a single instruction $c_i$ from the instruction set $I$ (Definition 3.1), takes the input values $in_i \in \mathcal{I}_i$, and produce the output values $out_i \in \mathcal{O}_i$. It is important to note that the output of $f_i$ can include both data flow elements (e.g., variables or memory locations with values assigned by $f_i$) and control flow elements (e.g., addresses of next interpreter functions $f_j \in f$ assigned by $f_i$). Consequently, we can express $f$ as the composition of different individual interpreters, i.e., $f_n \circ ... \circ f_1$, where later instructions act on the output of previous instructions.

**Program interpretation graph ($\mathcal{IG}$).** Programs often involve different control flow paths, such as if-else statements, leading compositions between individual interpreter functions to a directed graph instead of a linear sequence. This graph is referred to as the program interpretation graph. For a given CRU $c$, there can be multiple execution paths, each exercising different subsets of $\{f_1, ..., f_n\}$.

To construct the interpretation graph $\mathcal{IG} = (V, E)$, we consider all feasible execution paths of $c$. In $\mathcal{IG}$, each node $V_i \in V$ corresponds to $f_i$, and each directed edge $E_{i,j} \in E$ (connecting $V_i$ to $V_j$) represents at least one execution path where $f_j$ takes the output of $f_i$ as input, i.e., $E_{i,j} = (out_i, in_j)$.

**Automorphism group of interpretation graph.** Our objective is to find a group of symmetries that act on $c$ while preserving its input and output behavior as interpreted by $f$ in terms of $\mathcal{I}$ and $\mathcal{O}$ (Definition 3.6). Intuitively, as $\mathcal{IG}$ represents all execution paths of $c$, any transformations that preserve $\mathcal{IG}$ should also preserve the execution behavior of $c$. Therefore, we aim to uncover a group of symmetries that preserve $\mathcal{IG}$ (Theorem 1), and such a group can guide us to construct code analysis model that can stay invariant to all symmetries of the group (§3.4).

To achieve this, we consider a specific set of symmetries called the *automorphisms* of $\mathcal{IG}$, denoted as $Aut(\mathcal{IG})$. An automorphism is a group of symmetries $\sigma \in Aut(\mathcal{IG})$ that act on the interpretation graph $\mathcal{IG} = (V, E)$. Intuitively, graph automorphisms can be thought of as permutations of nodes that do not change the connectivity of the graph. $Aut(\mathcal{IG})$ is formally defined as follows:

**Definition 3.8** ($\mathcal{IG}$ Automorphism). $\mathcal{IG}$ automorphism is a group of symmetries $\sigma \in Aut(\mathcal{IG})$ acting on an interpretation graph $\mathcal{IG} = (V, E)$, where $\sigma$ is a bijective mapping: $\sigma : V \to V$, such that for every edge $E_{i,j} \in E$, i.e., connecting $f_i$ and $f_j$, there is a corresponding edge $(\sigma(f_i), \sigma(f_j)) \in E$.

We now show how the automorphism $\sigma \in Aut(\mathcal{IG})$ preserves all input and output behavior of $\{f_1, ..., f_n\}$ in the space of $\mathcal{I}$ and $\mathcal{O}$. As mentioned earlier, graph automorphism is a permutation on the set of nodes in $\mathcal{IG}$ such that the edges $E_{i,j} = (out_i, in_j)$ are preserved in the transformed $\mathcal{IG}'$. As each $f_i \in \{f_1, .., f_n\}$ operates on $c_i \in c$, we have the following (see Appendix B for the proof):

**Theorem 1.** The set of automorphisms $\sigma \in Aut(\mathcal{IG})$ forms a program symmetry group.

### 3.4  $Aut(\mathcal{IG})$-Equivariant Code Representation

Existing approaches for code analysis using Transformer typically involve an embedding layer followed by applying $l$ self-attention layers $A^l$. For downstream code analysis, a prediction head $F$, is placed on top of $A^l$. We can thus consider the representation learning $r$ as the composition of embedding layer and $A^l$, while the prediction head $F$ as the predictive learning $p$ (§3.1). We now present the development of a new self-attention layer that is $Aut(\mathcal{IG})$-equivariant.

**Self-attention.** The standard self-attention computation can be succinctly represented as $w_v \cdot s(w_k^T \cdot w_q)$, where $w_v$, $w_k$, and $w_q$ are learnable parameters for transforming value, key, and query, respectively, and $s(\cdot)$ represents scaling by $\sqrt{d}$ and applying Softmax (see Appendix A).

It is easy to show that the existing self-attention layer is equivariant to permutations (Appendix B). However, we want to make the self-attention layers equivariant *only* to $Aut(\mathcal{IG})$, not *all permutations*. In the following, we describe how to build $Aut(\mathcal{IG})$-equivariant self-attention.

**Biasing self-attention with a distance matrix.** To build $Aut(\mathcal{IG})$-equivariant self-attention layers, denoted as $GA$, we add a customized distance matrix $d_{\mathcal{IG}}$ to $GA$: $GA(e) = w_v e \cdot (s(w_k e^T \circ w_q e) + d_{\mathcal{IG}})$. Importantly, $d_{\mathcal{IG}}$ should have two properties: (1) $d_{\mathcal{IG}}$ stays invariant when $\sigma \in Aut(\mathcal{IG})$ acts on $\mathcal{IG}$: $d_{\mathcal{IG}} = \sigma(d_{\mathcal{IG}})$, and (2) $d_{\mathcal{IG}}$ commutes with permutation matrix $p_\sigma$ ($\sigma \in Aut(\mathcal{IG})$).

We will describe a concrete instantiation of $d_{\mathcal{IG}}$ in §4.2. Based on the two properties, we have the following Theorem (see Appendix B for the proof).

**Theorem 2.** Self-attention $GA(e) = w_v e \cdot (s(w_k e^T \cdot w_q e) + d_{\mathcal{IG}})$ is $Aut(\mathcal{IG})$-equivariant.

As the embedding layer is trivially permutation equivariant, composing it with $Aut(\mathcal{IG})$-equivariant self-attention layers leads to $Aut(\mathcal{IG})$-equivariant code representation learning (Lemma 1).

### 3.5  $Aut(\mathcal{IG})$-Invariant Predictor

We describe two prediction modules that are inherently $G$-invariant, so stacking them on top of the $Aut(\mathcal{IG})$-equivariant module leads to an $Aut(\mathcal{IG})$-invariant code model (Lemma 2).

**Token-level.** Token-level predictor is often employed when each input token needs a label, e.g., predicting memory region per instruction (§5). As the automorphism acts on the input sequence $e$ but not individual tokens, i.e., the value of the embedding vectors, the automorphism $\sigma$ does not apply to the query vector $q_i$ (§3.4). Therefore, we have Lemma 3. See Appendix B for complete proof.

**Lemma 3.** The biased self-attention computing the embedding $e_i' = GA(e_i)$ is $Aut(\mathcal{IG})$-invariant.

**Pooling-based.** Another popular $Aut(\mathcal{IG})$-invariant predictor involves pooling the embedding sequence $e' = GA(e)$, e.g., using max or mean. Pooling operators are invariant to permutations, thus to $Aut(\mathcal{IG})$, e.g., the mean pooling $\mu(e') = (\Sigma_{i=1}^n e_i')/n$ is not sensitive to the order of $(e_1', ..., e_n')$. Pooling-based predictor is often employed when we aim to predict the property for the entire input sequence, e.g., predicting the function signature, detecting function similarity, etc. (§5).

## 4  SymC Implementation

### 4.1  Relaxing $\mathcal{IG}$ to Program Dependence Graph

In §3.4, we demonstrated how to build $Aut(\mathcal{IG})$-equivariant self-attention layers. However, directly constructing $\mathcal{IG}$ is computationally impractical as we need to iterate all possible execution paths. To address this, we consider *program dependence graph* (PDG), a sound over-approximation to $\mathcal{IG}$ that explicitly captures the control/data dependencies and can be computed statically and efficiently.

PDG ($V_{PDG}, E_{PDG}$) is a super graph of $\mathcal{IG}$, sharing the same vertices but having a superset of edges ($E_{PDG} \supseteq E_{\mathcal{IG}}$), because we consider all memory accesses as aliasing, making PDG a conservative construction of $\mathcal{IG}$. Enforcing PDG to be a super graph of $\mathcal{IG}$ is crucial because the automorphism group of a subgraph is a subgroup of that of the super graph ($Aut(PDG) \supseteq Aut(\mathcal{IG})$). Thus, if the self-attention layer is $Aut(PDG)$-equivariant, it is guaranteed to be $Aut(\mathcal{IG})$-equivariant.

**PDG construction.** We construct PDG edges based on data/control dependencies between instructions. Three types of data dependencies (read-after-write, write-after-read, and write-after-write) are

considered, indicating the presence of data flow. Control dependencies are included to determine the execution order. These dependencies establish a partial order of instructions, preventing permutations that violate edge directions that might alter the input-output behavior of the program (§3.3).

## 4.2 ENCODING GRAPH STRUCTURE

This section presents a concrete instance of the distance matrix defined on PDG, which enables us to prove $Aut(PDG)$-equivariance for the resulting self-attention layers.

**Distance matrix.** Let $d$ denote the distance matrix of PDG where $d_{ij}$ represents the distance between nodes $V_i$ and $V_j$. Each entry $d_{ij}$ is a 2-value tuple $(p_{ij}, n_{ij})$, indicating the shortest path from the lowest common ancestor of $V_i$ and $V_j$, denoted as $T_{ij}$, to $V_i$ and $V_j$, respectively.

We incorporate $d$ into the multi-head self-attention (MHA), ensuring $Aut(PDG)$-equivariance, and define specific modifications to the attention heads to handle positive and negative distances. Specifically, the first half of the attention heads $MHA^i(e)$, for $i \in [1, h/2]$, are combined with the matrix $dp$ formed by the positive distances in $d$ (denoted as $dp_{ij} = p_{ij}$). The second half of the attention heads $MHA^i(e)$, for $i \in [h/2 + 1, h]$, are combined with the matrix $dn$ formed by the negative distances in $d$ (denoted as $dn_{ij} = n_{ij}$). The modified attention heads are defined as: (1) $MHA^i(e) = w_v e \cdot (s(w_k e^T \cdot w_q e) + dp), i \in [1, h/2]$, (2) $MHA^i(e) = w_v e \cdot (s(w_k e^T \cdot w_q e) + dn), i \in [h/2+1, h]$.

It is easy to show $d$ satisfies the two properties defined in §3.4 (see Appendix B). We thus have:

**Lemma 4.** The distance matrix $d$ of PDG remains invariant under the action of $\sigma \in Aut(PDG)$.

**Lemma 5.** The distance matrix $d$ of PDG commutes with permutation matrix $p_\sigma$ of the automorphism $\sigma \in Aut(PDG)$: $d \cdot p_\sigma = p_\sigma \cdot d$.

Based on these two properties, we can prove each head in MHA is $Aut(PDG)$-equivariant, following the same proof steps to Theorem 2. Therefore, according to Lemma 1, MHA composed by multiple $Aut(PDG)$-equivariant heads is also $Aut(PDG)$-equivariant.

## 5 EXPERIMENTAL SETUP

**Program analysis tasks.** We consider analysis tasks that require a deep understanding of code behavior such that they are expected to stay *invariant* to code symmetries. Specifically, we consider (1) *function name prediction*, which performs an "extreme summarization" of the function behavior. (2) *function similarity detection*, which predicts if a pair of functions are semantically similar; (3) *function signature prediction*, which predicts the types (int, float, etc.) of function arguments; and (4) *memory region prediction*, which predicts the memory region (stack, heap, etc.) that each memory-accessing instruction can possibly access. For (2)-(4), we focus on analyzing stripped binaries considering its broad applications, e.g., vulnerability detection and security retrofitting.

**Baselines.** We consider five baselines for function name prediction: code2vec (Alon et al., 2019), code2seq (Alon et al., 2018), GGNN (Fernandes et al., 2018), and two more general-purpose LLMs, GPT-4 and WizardCoder (3B) (Luo et al., 2023). Note that the dataset used to train LLMs might overlap with our test set. For example, Hadoop (our test set for function name prediction, see Appendix D) is included in BigCode (HuggingFace & ServiceNow, 2022), one of the widely used datasets to train code LLMs. Our goal is to demonstrate SYMC still generalizes better than existing code LLMs under such a disadvantaged setting.

For tasks (2)-(4), we compare to PalmTree (Li et al., 2021), the state-of-the-art binary analysis tool that covers all our considered tasks. To ensure a fair comparison, we include three PalmTree versions: PalmTree, PalmTree-O, and PalmTree-N. PalmTree is pre-trained on *2.25 billion* instructions. PalmTree-O is pre-trained on *137.6 million* instructions using our own dataset (Appendix D), with full access to fine-tuning and evaluation data (excluding labels), while not accessible by SYMC as it is not pre-trained. We aim to show SYMC's strong generalizability even in this disadvantaged setting. PalmTree-N serves as the baseline Transformer encoder without being pre-trained.

**Code transformations.** We consider a set of real-world semantics-preserving transformations beyond PDG automorphisms to evaluate SYMC's generalization by staying $Aut(PDG)$-equivariant. Instruction permutation occasionally forms the basis for these transformations. In particular, we

Table 2: Evaluation on samples under different percentages of semantics-preserving permutations. F1 measures the prediction performance of function name, function signature, and memory region. AUC (area under the ROC curve) measures the function similarity detection performance. The violation rate is highlighted in red . The larger the violation rate, the darker the color.

| | | Model Size | Train Size | F1 & AUC | | | | | Invariance Violation (%) | | | |
|---|---|---|---|---|---|---|---|---|---|---|---|---|
| | | | | 0% | 25% | 50% | 75% | 100% | 25% | 50% | 75% | 100% |
| Function Name | SYMC | 68.4M | 202M | 0.363 | 0.364* | 0.363 | 0.363 | 0.363 | 0 | 0.1* | 0 | 0 |
| | code2seq | 6.3M | 5.1G | 0.255 | 0.238 | 0.236 | 0.237 | 0.247 | 54 | 53 | 57 | 61 |
| | code2vec | 348M | 32G | 0.177 | 0.199 | 0.195 | 0.197 | 0.196 | 53 | 53 | 52 | 52 |
| | GGNN | 53M | 2.4G | 0.016 | 0.016 | 0.016 | 0.016 | 0.016 | 4 | 4 | 5 | 7 |
| | GPT-4 | N/A | N/A | 0.303 | 0.313 | 0.317 | 0.329 | 0.307 | 42 | 43 | 45 | 43 |
| | WizardCoder | 3B | N/A | 0.339 | 0.347 | 0.348 | 0.359 | 0.346 | 6 | 7 | 12 | 14 |
| Function Signature | SYMC | 58.3M | 12M | 0.88 | 0.88 | 0.88 | 0.88 | 0.88 | 0 | 0 | 0 | 0 |
| | PalmTree | 3.2M | 17.4G | 0.59 | 0.55 | 0.49 | 0.42 | 0.41 | 12 | 23 | 18 | 24 |
| | PalmTree-O | 3.2M | 5.3G | 0.49 | 0.48 | 0.45 | 0.41 | 0.41 | 19 | 6 | 12 | 6 |
| | PalmTree-N | 3.2M | 614M | 0.19 | 0.41 | 0.41 | 0.41 | 0.41 | 83 | 82 | 83 | 86 |
| Memory Region | SYMC | 58.9M | 340M | 0.86 | 0.86 | 0.86 | 0.86 | 0.86 | 0 | 0 | 0 | 0 |
| | PalmTree | 3.07M | 17.9G | 0.57 | 0.45 | 0.45 | 0.48 | 0.43 | 17 | 17 | 28 | 18 |
| | PalmTree-O | 3.07M | 5.8G | 0.57 | 0.42 | 0.45 | 0.47 | 0.44 | 10 | 13 | 14 | 11 |
| | PalmTree-N | 3.07M | 1.1G | 0.32 | 0.22 | 0.29 | 0.17 | 0.2 | 30 | 36 | 31 | 32 |
| Function Similarity | SYMC | 58.9M | 133M | 0.96 | 0.96 | 0.96 | 0.96 | 0.96 | 0 | 0 | 0 | 0 |
| | PalmTree | 3.06M | 17.4G | 0.72 | 0.61 | 0.53 | 0.71 | 0.69 | 18 | 19 | 30 | 31 |
| | PalmTree-O | 3.06M | 5.3G | 0.8 | 0.79 | 0.76 | 0.72 | 0.72 | 30 | 28 | 30 | 35 |
| | PalmTree-N | 3.06M | 614M | 0.71 | 0.64 | 0.56 | 0.66 | 0.72 | 11 | 18 | 24 | 38 |

*We observe a slight value change due to the floating point precision error by adopting memory-efficient 16-bit.

consider two categories of binary code transformations: (1) *compiler optimizations* from GCC-7.5 and Clang-8, some of which reorder instructions for scheduling purposes (`-fdelayed-branch`, `-fschedule-insns`); and (2) *compiler-based obfuscations*, where we consider 5 obfuscations following Jin et al. (2022), such as control flow flattening, indirect branching, etc.

In addition to binary transformations, we consider six source code transformations following Rabin et al. (2021) and Wang et al. (2022): *variable rename* - changes the name of the identifiers; *statement permute* – permutes statements but preserves the code blocks's input-output behavior; *loop exchange* – transforms `for` from/to `while` loops; *boolean exchange* – flips the boolean variables and negates all their uses by tracking their def-use chain; *unused statement* – injects unused string declaration into a randomly chosen basic block; *switch to if* – transforms `switch` from/to `if` statements.

## 6 EVALUATION

### 6.1 INVARIANCE AND GENERALIZATION

**Evaluating $Aut(PDG)$-invariance.** As SYMC is provably invariant to $Aut(PG)$, we aim to study how other baselines perform under varying percentages of semantics-preserving statement permutations. Table 2 shows that all baselines, even some of which have much larger model sizes and are extensively pre-trained with samples that potentially include the test set (§5), are susceptible to slight permutations (e.g., 25%), i.e., with their prediction changed by 27.8% on average.

**Generalization to other semantics-preserving transformations.** Table 3 shows that SYMC generalizes to new samples transformed by unseen semantics-preserving transformations that are not part of $Aut(PDG)$, outperforming the second-best non-LLM based approach, code2seq, by 30.8%. It also outperforms the two LLMs, GPT-4 and WizardCoder, by 16.1% and 1.63%, respectively. We integrate CodeWordNet (Jin et al., 2022) to relax predicted names to a cluster of synonyms, addressing the issue of ambiguity of function names. However, the performance of SYMC decreases to 0.309 (was 0.374) when we measure the exact match. This performance gap shows that large pre-trained code models possess a more comprehensive understanding of natural language, especially beneficial for

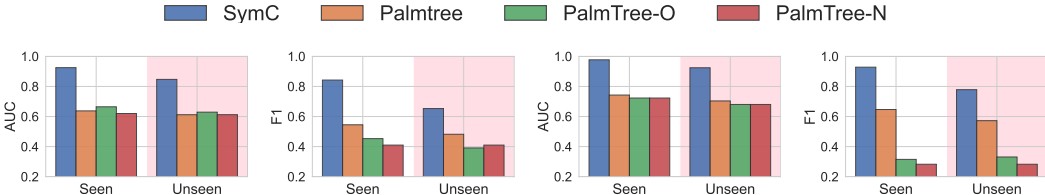

Figure 2: Evaluation on unseen optimization and obfuscation (marked in pink). We also include the testing results on seen optimizations and obfuscations.

function name prediction. Nonetheless, SYMC retains the edge of having a much smaller model size (108×), eliminating the need for extensive pre-training, and offering provable robustness.

Besides basic source rewrites, we compare SYMC to baselines on generalization to unseen compiler optimizations and obfuscations. Figure 2 shows that SYMC outperforms PalmTree across all binary analysis tasks (we do not include memory region prediction as the dataset does not have such information (Guo et al., 2019)) by 33.8% and 30.7% on seen and unseen transformations, respectively. While the compiler optimizations and obfuscations often involve more sophisticated transformations not directly related to instruction permutations, SYMC maintains its superior generalization.

## 6.2 TRAINING EFFICIENCY

Besides the improved robustness and generalization, SYMC is efficient in training in avoiding expensive pre-training efforts, e.g., some may take up to 10 days (Jin et al., 2022). As shown in §6.1, SYMC, without any pre-training, outperforms the pre-trained baselines.

In this section, we assess SYMC's performance under the limited training resources and how much training effort it can save. Specifically, we reduce the *model sizes* and *training iterations* to test the

Table 3: The performance (F1) of SYMC and baselines against different unseen code transformations.

|  |  | SYMC | code2seq | code2vec | GGNN | GPT-4 | WizardCoder |
|---|---|---|---|---|---|---|---|
| Variable | 0% | 0.389 | 0.334 | 0.264 | 0.029 | 0.356 | 0.362 |
| Rename | 100% | 0.375 | 0.335 | 0.247 | 0.026 | 0.351 | 0.361 |
| Statement | 0% | 0.363 | 0.241 | 0.177 | 0.019 | 0.303 | 0.339 |
| Permute | 100% | 0.363 | 0.234 | 0.196 | 0.019 | 0.307 | 0.346 |
| Loop | 0% | 0.373 | 0.283 | 0.243 | 0.007 | 0.310 | 0.379 |
| Exchange | 100% | 0.357 | 0.299 | 0.241 | 0.007 | 0.308 | 0.366 |
| Boolean | 0% | 0.421 | 0.332 | 0.268 | 0.031 | 0.329 | 0.414 |
| Exchange | 100% | 0.412 | 0.272 | 0.242 | 0.026 | 0.323 | 0.406 |
| Unused | 0% | 0.347 | 0.296 | 0.267 | 0.016 | 0.316 | 0.358 |
| Statement | 100% | 0.342 | 0.285 | 0.26 | 0.012 | 0.309 | 0.350 |
| Switch | 0% | 0.372 | 0.31 | 0.376 | 0.027 | 0.326 | 0.385 |
| to If | 100% | 0.372 | 0.293 | 0.33 | 0.009 | 0.332 | 0.379 |

hypothesis that SYMC requires less training effort for similar testing performance due to its improved training efficiency. Figure 3 shows that SYMC's performance (on memory region prediction) remains the highest in both reduced size and training iterations, outperforming PalmTree by 36.9% and 21.4%, respectively. Even in the most strict scenario, SYMC remains 38.2% and 15.3% better in both settings.

We then study the training effort (including both pre-training and fine-tuning) of SYMC and PalmTree. Table 4 shows their GPU hours, power, and emitted carbon dioxide estimation when they reach 0.5 F1 score in memory region prediction. We assume the GPU always reaches its power cap (350W) to estimate an upper bound of the power usage. $CO_2$eq stands for the carbon dioxide equivalent, a unit for measuring carbon footprints. By being more training efficient, SYMC incurs 1,281× less total GPU time, power, and emitted carbon dioxide than PalmTree in obtaining the same performance.

## 6.3 ABLATIONS OF DESIGN CHOICES

**Equivariance vs. Invariance.** We compare the $Aut(PDG)$-*equivariant* self-attention layers to the $Aut(PDG)$-*invariant* ones, an alternative design choice to implement $Aut(PDG)$-*invariant* code

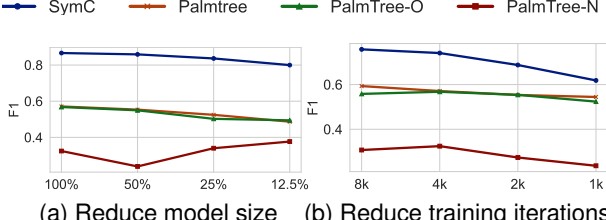

Table 4: The resource consumed by training SYMC and other baselines to reach 0.5 F1 score in memory region prediction.

|  | Time (Hours) | Power (kWh) | Carbon (CO$_2$eq) |
|---|---|---|---|
| SYMC | 0.07 | 0.025 | 0.009 |
| PalmTree-O[*] | 89.67 | 31.38 | 11.64 |

[*]PalmTree did not disclose its hours for pre-training, so we include the pre-training time (in 10 epochs) based on our own pre-trained PalmTree.

Figure 3: Comparing SYMC and baselines on constrained resources, where we (a) reduce the model weights, and (b) reduce the number of training iterations, and observe how that affects the performance.

models. Figure 4a shows that setting layers invariant early hinders prediction performance. SYMC with equivariant layers has an average 0.73 F1 across all training iterations and outperforms the second-best setting by 60.7%. This observation confirms the empirical findings that making earlier layers equivariant instead of invariant leads to better performance (Higgins et al., 2018).

**Adding pre-training.** We explore the impact of pre-training SYMC with masked language modeling (Devlin et al., 2018). We compare SYMC (without pre-training by default) to pretrained versions with varying pretraining iterations. We then finetune them for memory region prediction. Figure 4b shows that pretraining with even one epoch results in a significantly improved F1 score, e.g., by 10.8%, with much faster convergence. How-

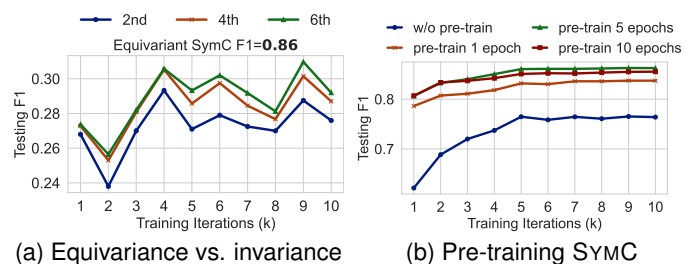

Figure 4: (a) Comparing SYMC's equivariant self-attention layers with setting them *invariant* starting at earlier layer. (b) Comparing SYMC when adding pre-training with varying pre-training epochs.

ever, additional pre-training epochs show diminishing returns, likely due to the limited training samples, e.g., the F1 score only improves by 3.2% with pre-training five epochs compared to 1 epoch.

## 7  RELATED WORK

**Code representation learning.** Previous research aims to automate software development tasks through code representation learning (Ding et al., 2023; Feng et al., 2020; Guo et al., 2022; Ahmad et al., 2021), employing new architectures and pre-training objectives (Hellendoorn et al., 2019; Allamanis et al., 2017; Sun et al., 2020; Peng et al., 2021; Kim et al., 2021; Guo et al., 2020). However, unlike our approach, these approaches cannot provide any guarantees of encoding semantics.

**Symmetry in machine learning.** Symmetry plays a crucial role in creating efficient neural architectures across various domains (Reiser et al., 2022; Wang et al., 2020; Bogatskiy et al., 2020; Perraudin et al., 2019; Cohen & Welling, 2016; Gordon et al., 2019; Dehmamy et al., 2021). Different architectures, such as CNNs, GNNs, and Transformers, leverage symmetry to handle translations, rotations, permutations, etc. (Lee et al., 2019; Cohen & Welling, 2016; Esteves et al., 2018; Hutchinson et al., 2021; Gordon et al., 2019; Romero & Cordonnier, 2020). SYMC sets the first step to formalize code semantics learning using symmetry groups.

## 8  CONCLUSION

We studied code symmetries' impact on code LLM architectures for program reasoning tasks, introducing a novel self-attention variant that brought significant gains in generalization and robustness across a variety of program analysis tasks, providing valuable insights for specialized LLM development in reasoning and analyzing programs.

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

## A  PRELIMINARIES

This section formally defines the symmetry group and the invariance and equivariance properties against the symmetry group.

**Symmetry group.** Intuitively, a *symmetry* is a transformation or operation on an object that preserves certain properties of the object. For example, in the context of image classification, a rotation operation acting on an image of a ball, which does not change the label of the ball, can be considered a symmetry. A symmetry group is a set of such symmetries with some additional properties. An arbitrary set of symmetries does not always form a symmetry group. To form a symmetry group, a set of operations must possess certain additional properties as described below.

**Definition A.1.** A *symmetry group* $(G, \circ)$ consists of a non-empty set $G$ of transformations and a binary operator $\circ : G \times G \to G$, where $\circ$ operates on two elements (i.e., transformations) in $G$, e.g., $x, y \in G$, and produces a new transformation $z = x \circ y$. $(G, \circ)$ should satisfy four axioms:

- **Associativity**: $\forall x, y, z \in G, x \circ (y \circ z) = (x \circ y) \circ z$

- **Identity**: $\exists 1 \in G, \forall x \in G, x \circ 1 = 1 \circ x$

- **Inverse**: $\forall x \in G, \exists x^{-1} \in G, x \circ x^{-1} = 1$

- **Closure**: $\forall x, y \in G, x \circ y \in G$

**Action of a symmetry group.** As defined above, the elements of a $G$ are abstract transformations that become concrete when they *act* on some set $X$, i.e., they transform some object $x \in X$ into another object $x' \in X$ while keeping some properties of the object *invariant*. Formally, an action of a symmetry group $G$ is defined as follows:

**Definition A.2.** An **action** $\bullet$ of a symmetry group $(G, \circ)$ is a binary operation defined on a set of objects $X$, i.e., $\bullet : G \times X \to X$,[1] where

- **Identitiy:** $\forall x \in X, 1 \bullet x = x$

- **Compatibility:** $\forall g, h \in G, x \in X, (g \circ h) \bullet x = g \bullet (h \bullet x)$

As a concrete example, $X$ can be a set of programs and $G$ can be all possible instruction permutations that preserve the input-output behavior of the programs in $X$. It might seem unclear at this point how these permutations form a group (satisfying group axioms). We will formalize the notion of permutations and their actions on programs in §3.3.

**Notation.** It is common in the group theory literature to use $\circ$ to denote both *action* and *composition*, when it is clear from the context which operation is being used (Higgins et al., 2018). For example, $(g \circ h) \circ x$ denotes *composing* the two transformations $g$ and $h$ and then letting the composite transformation *act* on an object $x$. It is also customary to interchange $g(x)$ and $g \circ x$ where both denote applying a function/action on $x$. Therefore, we treat $g \bullet (h \bullet x)$, $g \circ (h \circ x)$, and $g(h(x))$ as the same and follow this convention in the rest of this paper.

**Invariance and equivariance.** A symmetry group comes with two properties, namely *invariance* and *equivariance*, that formalize the concept of preservation of some properties when a set $X$ is acted upon by the symmetry group $G$. Invariance refers to the property that remains unchanged under the action of the symmetry group. Equivariance, on the other hand, expresses the compatibility between the action of the symmetry group and the property.

To define this more precisely, we need to introduce a function $f : X \to Y$, where $X$ is the set under consideration and $Y$ is the co-domain representing the range of possible values associated with the property of interest. The function $f$ maps each element $x \in X$ to a corresponding element $y$ in the set $Y$, indicating the property's value for that particular element. We now define the equivariance and invariance of $f$ operating on $X$ against the group operations in $G$.

**Definition A.3.** Let $f : X \to Y$ be a function where $X$ and $Y$ are two sets and $G$ be the symmetry group that acts on both sets $X$ and $Y$.[2]

- **Invariant**: $f$ is called $G$-**invariant** if $\forall g \in G, \forall x \in X, f(g \circ x) = f(x)$.

- **Equivariant**: $f$ is called $G$-**equivariant** if $\forall g \in G, \forall x \in X, f(g \circ x) = g \circ f(x)$.

**Self-attention layers.** Given the embeddings of all vertices $f_i$ from $\mathcal{IG}$, we consider a sequence of embeddings by flattening $\mathcal{IG}$ following the order of instructions in $c$. Let this sequence of embeddings be denoted as $e = (e_1, ..., e_n)$. The self-attention computation, denoted as $A$, takes $e$ as input and produces another sequence of embeddings, denoted as $(e'_1, ..., e'_n)$.

---

[1] In group theory literature, this is often called the left action, but we will omit "left" as it is the only type of action we will use in this paper.

[2] We assume that $X$ and $Y$ have the same number of elements for the action of $G$ to be defined on both $X$ and $Y$.

The core operations in self-attention $A$ involve updating each embedding $e_i$ through the following steps:

1. First, it maps each embedding $e_i$ to three embeddings (query, key, and value): $q_i = f_q(e_i)$, $k_i = f_k(e_i)$, $v_i = f_v(e_i)$, where $f_q$, $f_k$, and $f_v$ are affine transformations (i.e., fully-connected linear layers) parameterized by $w_q$, $w_k$, and $w_v$, respectively.

2. Next, it computes the attention score $a_{ij}$ between every pair of embeddings $e_i$ and $e_j$ by taking the dot product between the query $q_i$ of $e_i$ and the key $k_j$ of $e_j$: $a_{ij} = q_i \cdot k_j$. The attention scores form a square matrix, where each cell $a_{ij}$ indicates the attention that $e_i$ should pay to $e_j$. The attention scores are then divided by $\sqrt{d}$ (the dimension of the embedding vectors), scaled using the softmax function to ensure they sum up to 1: $\hat{a}_{ij} = \frac{\exp(a_{ij})}{\sum_{j=1}^{n} \exp(a_{ij})}$. These two operations are denoted by $s$.

3. Finally, the scaled attention score $\hat{a}_{ij}$ is multiplied by $v_j$, and a vector sum is computed: $e_i' = \sum_{j=1}^{n} \hat{a}_{ij} v_{ij}$.

## B  PROOFS

**Lemma 1.** Let $f_1$ and $f_2$ be two functions that are both $G$-equivariant and $h = f_1 \circ f_2$ be the new function composed by $f_1$ and $f_2$. $h$ is also $G$-equivariant.

*Proof.* For all $g \in G$ and any input $x$, we have

$$
\begin{aligned}
h(g \circ x) &= (f_1 \circ f_2)(g \circ x) \\
&= f_1(f_2(g \circ x)) && \triangleright \text{Associativity} \\
&= f_1(g \circ f_2(x)) && \triangleright f_2 \text{ is equivariant to } g \\
&= g \circ f_1(f_2(x)) && \triangleright f_1 \text{ is equivariant to } g \\
&= g \circ (f_1 \circ f_2)(x) && \triangleright \text{Associativity} \\
&= g \circ h(x)
\end{aligned}
$$

Therefore, $h(g \circ x) = g \circ h(x)$, so $h$ is $G$-equivariant.

**Lemma 2.** Let $f_1$ and $f_2$ be two functions where $f_1$ is $G$-equivariant $f_2$ is $G$-invariant, and $h = f_2 \circ f_1$ be the new function composed by applying $f_1$ and then $f_2$. $h$ is $G$-invariant.

*Proof.* For all $g \in G$ and any input $x$, we have

$$
\begin{aligned}
h(g \circ x) &= (f_2 \circ f_1)(g \circ x) \\
&= f_2(f_1(g \circ x)) && \triangleright \text{Associativity} \\
&= f_2(g \circ f_1(x)) && \triangleright f_1 \text{ is equivariant to } g \\
&= f_2(f_1(x)) && \triangleright f_2 \text{ is invariant to } g \\
&= (f_2 \circ f_1)(x) && \triangleright \text{Associativity} \\
&= h(x)
\end{aligned}
$$

**Theorem 1.** The set of automorphisms $\sigma \in Aut(\mathcal{IG})$ forms a program symmetry group.

*Proof.* Consider an arbitrary $\sigma \in Aut(\mathcal{IG})$. Definition 3.8 states that for all $f_i \in \{f_1, ..., f_n\}$, $\sigma(f_i)$ have the same edges as $\mathcal{IG}$ before $\sigma$ was applied. As $\sigma$ is a permutation and there is also a bijective mapping between $f_i$ and $c_i$, i.e., $f_i$ always interprets $c_i$, we have $\sigma(f_i) = f_i(\sigma \circ c_i, in_i)$. Definition 3.8 also states that $\sigma(f_i)$ is connected with the same edges. Therefore, the output of $\sigma(f_i) = out_i$. We thus have $f_i(\sigma \circ c_i, in_i) = out_i = f_i(c_i, in_i), \forall \sigma \in Aut(\mathcal{IG})$ and $\forall f_i \in \{f_1, ..., f_n\}$. Therefore, all $\sigma \in Aut(\mathcal{IG})$ are semantics-preserving program symmetries, according to Definition 3.6. Moreover, it is well known in the literature that the automorphisms of any graph form a group by satisfying group axioms (Definition A.1) (Biggs et al., 1993; West et al., 2001). Therefore, $Aut(\mathcal{IG})$ forms a group of program symmetries, according to Definition 3.7: $Aut(\mathcal{IG}) \in G$.

**Permutation matrix.** Let $\pi$ be a symmetry in the permutation group that permutes input embeddings $e \in \mathbb{R}^{d \times n}$ to the self-attention layer. Applying $\pi$ is done by $e$ with a permutation matrix $p_\pi \in \{0, 1\}^{n \times n}$ (Knuth, 1970). $p_\pi$ is an orthogonal binary matrix with a single 1 in each column and row, and 0s elsewhere. Right-multiplying $e$ with $p_\pi$ permutes columns, and left-multiplying $e^T$ with $p_\pi^T$ permutes rows.

**Theorem 2.** The biased self-attention layer, $GA(e) = w_v e \cdot (s(w_k e^T \cdot w_q e) + d_{\mathcal{IG}})$, is $Aut(\mathcal{IG})$-equivariant.

*Proof.*

$$GA(\sigma \cdot e)$$
$$= w_v \sigma(e) \cdot (s(w_k \sigma(e)^T \cdot w_q \sigma(e)) + \sigma(d_{\mathcal{IG}}))$$

$\sigma(\cdot)$ denotes applying the permutation matrix $p_\sigma$. As we have $\sigma(d_{\mathcal{IG}}) = d_{\mathcal{IG}}$ (the first property of $d_{\mathcal{IG}}$):

$$= w_v e p_\sigma \cdot (s((w_k e p_\sigma)^T \cdot w_q e p_\sigma) + d_{\mathcal{IG}})$$
$$= w_v e p_\sigma \cdot s(p_\sigma^T (w_k e)^T \cdot w_q e p_\sigma) + w_v e p_\sigma \cdot d_{\mathcal{IG}}$$

Softmax $s$ is permutation equivariant, and $d_{\mathcal{IG}} \cdot p_\sigma = p_\sigma \cdot d_{\mathcal{IG}}$ (the second property of $d_{\mathcal{IG}}$):

$$= w_v e(p_\sigma p_\sigma^T) \cdot s((w_k e)^T \cdot w_q e) \cdot p_\sigma + w_v e \cdot d_{\mathcal{IG}} \cdot p_\sigma$$
$$= w_v e \cdot ((s(w_k e)^T \cdot w_q e) + d_{\mathcal{IG}}) \cdot p_\sigma$$
$$= \sigma(GA(e))$$

**Lemma 3.** The biased self-attention layer computing the embedding $e_i' = GA(e_i)$ is $Aut(\mathcal{IG})$-invariant.

*Proof.*

$$e_i' = GA(\sigma \cdot e_i)$$
$$= w_v \sigma(e) \cdot (s(w_k \sigma(e)^T \cdot w_q e_i) + \sigma(d_i))$$

$d_i$ is a column vector, so permuting the row of $d_i$ is achieved by $p_\sigma^T d_i$ (see §3.4):

$$= w_v e p_\sigma \cdot (s((w_k e p_\sigma)^T \cdot w_q e_i) + p_\sigma^T d_i)$$
$$= w_v e p_\sigma \cdot s(p_\sigma^T (w_k e)^T \cdot w_q e_i) + w_v e p_\sigma \cdot p_\sigma^T d_i$$
$$= w_v e(p_\sigma p_\sigma^T) \cdot s((w_k e)^T \cdot w_q e_i) + w_v e \cdot (p_\sigma p_\sigma^T) \cdot d_i$$

$p_\sigma$ is an orthogonal matrix (see §3.4):

$$= w_v e \cdot ((s(w_k e)^T \cdot w_q e_i) + d_i)$$
$$= GA(e_i)$$

**Lemma 4.** The distance matrix $d$ of PDG remains invariant under the action of $\sigma \in Aut(PDG)$.

*Proof.* We need to show that the shortest path $p_{\sigma(i)\sigma(j)}$ from $\sigma(T_{ij})$ to $\sigma(V_i)$ remains the same as $p_{ij}$ (the same applies to $n_{\sigma(i)\sigma(j)}$). Without loss of generality, we focus on proving $p_{\sigma(i)\sigma(j)} = p_{ij}$.

Assume there exists a shortest path $P = (T_{ij}, ..., V_i)$. Let $P' = (\sigma(T_{ij}), ..., \sigma(V_i))$ be the corresponding shortest path in $\sigma(PDG)$ under the automorphism $\sigma$. We need to demonstrate two properties.

First, $P'$ is a valid path from $\sigma(T_{ij})$ to $\sigma(V_i)$. Since $P$ is a valid path, $T_{ij}$ is adjacent to its next node in $P$ (denoted as $V_m$), and this holds for every pair of neighboring nodes until $V_i$. As $\sigma$ is an

automorphism, the same adjacency relationship holds for $P'$, where $\sigma(T_{ij})$ is adjacent to $\sigma(V_m)$ and so on, until $\sigma(V_i)$. Hence, $P'$ is a valid path from $\sigma(T_{ij})$ to $\sigma(V_i)$ in PDG.

Second, we aim to show that $|P| = |P'|$, meaning $p_{\sigma(i)\sigma(j)} = p_{ij}$. Suppose, for contradiction, that $p_{\sigma(i)\sigma(j)} \neq p_{ij}$. Let's consider the case where $p_{\sigma(i)\sigma(j)} < p_{ij}$. This implies that the length of the path $P' = (\sigma(T_{ij}), \sigma(V_m), ..., \sigma(V_n), \sigma(V_i))$ is shorter than $p_{ij}$.

Now, let's apply $\sigma^{-1}$ to each node in $P'$, resulting in $\sigma^{-1}(P')$. Since $\sigma^{-1}$ is also in $Aut(PDG)$ and $\sigma^{-1}(\sigma(V)) = V$ (Definition A.1), each pair of adjacent nodes in $P'$, after applying $\sigma^{-1}$, remains adjacent. Furthermore, the path formed by these adjacent nodes has a length of $p_{\sigma(i)\sigma(j)}$, connecting $T_{ij}$ and $V_i$ in the original PDG.

Therefore, we obtain a path in PDG connecting $T_{ij}$ and $V_i$ that is shorter than $p_{ij}$, contradicting the fact that $p_{ij}$ is the shortest path in PDG between $T_{ij}$ and $V_i$. Thus, we reject the assumption that $p_{\sigma(i)\sigma(j)} < p_{ij}$.

Similarly, we can prove that $p_{\sigma(i)\sigma(j)} > p_{ij}$ is also false by demonstrating its contradiction with the fact that $p_{\sigma(i)\sigma(j)}$ is the shortest path in $\sigma(PDG)$.

Hence, we conclude that $p_{\sigma(i)\sigma(j)} = p_{ij}$, and as a result, the positive distance matrix $dp$ remains invariant under the action of $\sigma \in Aut(PDG)$.

By following the same steps, we can prove that $n_{\sigma(i)\sigma(j)} = n_{ij}$, demonstrating the invariance of the negative distance matrix $dn$ under the action of $\sigma \in Aut(PDG)$.

Therefore, the distance matrix $d$ remains invariant.

**Lemma 5.** The distance matrix $d$ of $PDG$ commutes with permutation matrix $p_\sigma$ of the automorphism $\sigma \in Aut(PDG)$: $d \cdot p_\sigma = p_\sigma \cdot d$.

*Proof.* According to Lemma 4, we have:

$$
\begin{aligned}
p_\sigma^T \cdot d \cdot p_\sigma &= d \\
p_\sigma \cdot p_\sigma^T \cdot d \cdot p_\sigma &= p_\sigma \cdot d \qquad &&\triangleright \text{Apply } p_\sigma \text{ on both side} \\
d \cdot p_\sigma &= p_\sigma \cdot d \qquad &&\triangleright p_\sigma \text{ is orthogonal matrix}
\end{aligned}
$$

**Lemma 6.** Standard self-attention layer $A$ is equivariant to the group of all permutations of input sequences.

*Proof.* Based on the operations performed by the self-attention layer and the permutation matrix, we can show the equivariance property as follows (Ji et al., 2019):

$$
\begin{aligned}
&A(\pi \cdot e) \\
&= w_v \pi(e) \cdot s(w_k \pi(e)^T \cdot w_q \pi(e)) \\
&= w_v e p_\pi \cdot s((w_k e p_\pi)^T \cdot w_q e p_\pi) \qquad &&\triangleright \text{Applying } p_\pi \\
&= w_v e p_\pi \cdot s(p_\pi^T (w_k e)^T \cdot w_q e p_\pi) \qquad &&\triangleright \text{Transpose of a product} \\
&= w_v e (p_\pi p_\pi^T) \cdot s((w_k e)^T \cdot w_q e) p_\pi \\
&= w_v e \cdot s((w_k e)^T \cdot w_q e) p_\pi \qquad &&\triangleright p_\pi \text{ is orthogonal matrix} \\
&= \pi(A(e))
\end{aligned}
$$

## C SYMC IMPLEMENTATION DETAILS

**Input sequences to self-attention.** The Transformer self-attention layer takes an input sequence of embeddings $e$ generated by the embedding layer $Emb$. It consists of four input sequences: the

instruction sequence $c$, per-instruction positional embeddings, and node centrality, denoted as $x_c$, $x_{pos}$, $x_{ind}$, and $x_{outd}$, respectively. For example, given the instruction sequence `a=a+1;b=a`, $x_c$ represents the tokenized sequence as (`a`,`=`,`a`,`+`,`1`,`b`,`=`,`a`). $x_{pos}$ assigns positions such that each new instruction/statement begins with position 1 of its first token and increases by 1 for each subsequent token within the instruction.

The centrality of each instruction is encoded by the in-degree and out-degree of the corresponding node in PDG. For each token in $c_i$, we annotate it with its in-degree (number of incoming edges) and out-degree (number of outgoing edges). For instance, in the case of `a=a+1;b=a`, the in-degree sequence $x_{ind}$ is $(0, 0, 0, 0, 0, 1, 1, 1)$, and the out-degree sequence $x_{outd}$ is $(1, 1, 1, 1, 1, 0, 0, 0)$.

We embed the four sequences independently using the embedding layers $Emb_c$, $Emb_{pos}$, $Emb_{ind}$, and $Emb_{outd}$. The final input embedding sequences $Emb(x)$ are obtained by summing the embedded sequences for each token: $Emb(x) = Emb_c(x_c) + Emb_{pos}(x_{pos}) + Emb_{ind}(x_{ind}) + Emb_{outd}(x_{outd})$. We have the following lemma:

**Lemma 7.** The sum of the input embedding tokens sequences is $Aut(PDG)$-equivariant: $Emb(\sigma \circ x) = \sigma \circ Emb(x)$.

Group axiom of inclusion specifies that composing the $Aut(PDG)$-equivariant embedding layers with $Aut(PDG)$-equivariant MHA layers results in an $Aut(PDG)$-equivariant representation learning component $r$ in our implementation.

## D  DETAILED EXPERIMENT SETUP

### D.1  IMPLEMENTATION DETAILS

We implement SYMC in 40,162 lines of code using Fairseq (Ott et al., 2019) PyTorch (Paszke et al., 2019). To compute PDG for x86 assembly code, we utilize Ghidra to lift the assembly code into P-Code, an intermediate representation used by Ghidra, to track implicit data and control flow via FLAGS register. To compute PDG for Java functions, we employ JavaParser on Java ASTs to analyze control and data dependencies. We conduct all the experiments on three Linux servers with Ubuntu 20.04 LTS, each featuring an AMD EPYC 7502 processor, 128 virtual cores, and 256GB RAM, with 12 Nvidia RTX 3090 GPUs in total.

**Datasets.** We use the Java dataset collected by Allamanis et al. (2016) to evaluate the function name prediction. The dataset includes 11 Java projects, such as Hadoop, Gradle, etc., totaling 707K methods and 5.6M statements. We fix Hadoop as our test set and use the other projects for training, to ensure the two sets do not overlap. For binary analysis, we collect and compile 27 open-source projects such as OpenSSL, which contains approximately 1.13M functions and 137.6M instructions.

### D.2  EXPERIMENT CONFIGURATIONS

**Hyperparameters.** We use SYMC with 8 attention layers, 12 attention heads, and a maximum input length of 512. For training, we use 10 epochs, a batch size of 64, and 14K/6K training/testing samples (strictly non-overlapping) unless stated otherwise. We employ 16-bit weight parameters for SYMC to optimize for memory efficiency.

**Evaluation metrics.** For most analysis tasks (§5), we use *F1 score*, the harmonic mean of precision and recall. For function similarity detection, we use Area Under Curve (AUC) of ROC curve, as it handles continuous similarity scores and varying thresholds for determining function similarity. We note that AUC-ROC might not be the most reliable metric (Arp et al., 2022), but we choose it primarily for comparing to the baselines whose results are measured in AUC-ROC (Li et al., 2021).

## E  MORE DETAILED EXPERIMENTS

**Unseen optimizations.** We vary the compiler optimizations in training and evaluation and include reference experiments where the training and evaluation share the same optimization options (marked in gray). For function similarity detection, training on `O0-O1` means the function pair has one function compiled with `O0` and the other with `O1`. In the case of evaluating on unseen optimizations, the

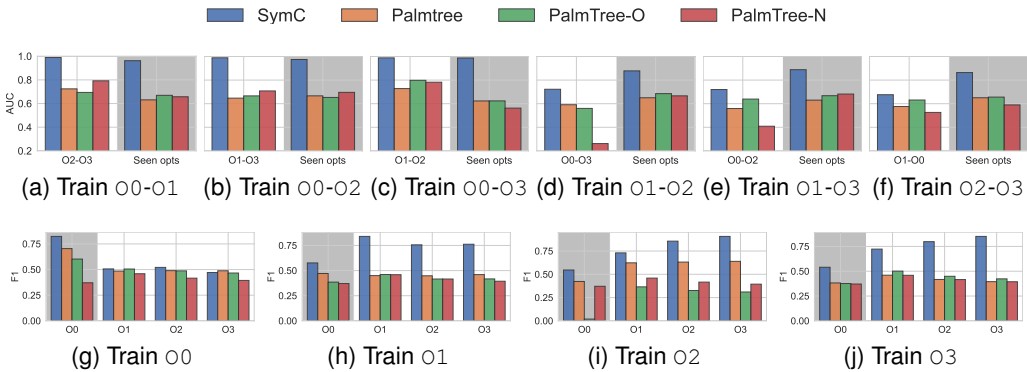

Figure 5: Unseen optimization evaluation. The upper row, i.e., (a)-(f), shows the results on function similarity detection. The lower row, i.e., (g)-(j), are results on function signature prediction. We also include the evaluation on seen optimizations (marked in gray).

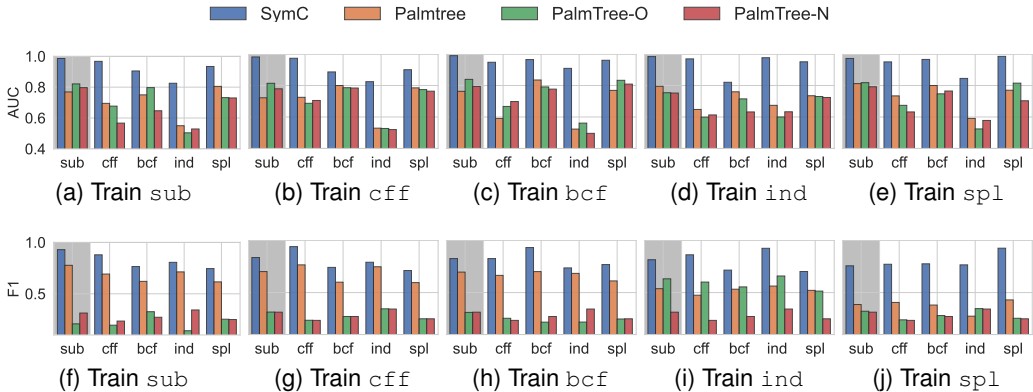

Figure 6: Unseen obfuscations evaluation. Similar to Figure 5, the upper row, i.e., (a)-(e), shows the results on function similarity detection. The lower row, i.e., (f)-(j), are results on function signature prediction. We also include the evaluation on seen optimizations (marked in gray).

corresponding testing set has to come from those compiled with O2-O3 to ensure the optimizations are unseen.

Figure 5 shows that SYMC outperforms PalmTree by 31% when evaluated on unseen optimizations. SYMC experiences a performance drop (e.g., by 28.6%) when not trained on O0 but tested on those compiled with O0. We believe this drop is caused by the extensive optimizations already enabled at the O1 (e.g., GCC employs 47 optimizations to aggressively reduce execution time and code size). The shift in distribution between O1 and O0 is much more pronounced than between O2 and O1, indicated by a KL divergence of 1.56 from O1 to O0 compared to 0.06 (96.2% lower) from O3 to O2. Nevertheless, when evaluated on seen optimizations, SYMC outperforms PalmTree by 28.1% on average.

**Unseen obfuscations.** We compare SYMC to baselines on generalization to unseen obfuscations. Figure 6 shows that SYMC outperforms PalmTree (on average) on unseen and seen obfuscations by 33.3% and 36.6%, respectively. Similar to the observations in evaluating unseen optimizations, while the obfuscations are not directly related to instruction permutations (i.e., automorphisms in $Aut(PDG)$), SYMC maintains its superior performance.

**Unseen lengths.** Besides the code transformations, we look into SYMC's generalization to *longer* sequences than those seen in training, a popular task for evaluating model generalizability (Gordon et al., 2019). We divide samples into four length bins (bin1 to bin4) based on their distribution in the dataset (§5). The bins are non-overlapping and increase in length. For example, we used bins

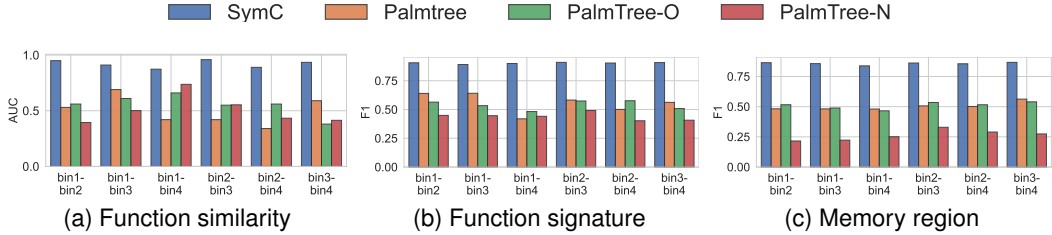

Figure 7: Evaluation on unseen samples with longer lengths. `bin1-bin4` denotes training on samples with lengths in `bin1` and testing on those in `bin4`.

[0-10], [1-20], [21-50], and [51-500] for function similarity detection. Figure 7 demonstrates that SYMC maintains strong generalization to longer sequences, outperforming PalmTree by 41.8%.

