# OpenReview forum: "Exploiting Code Symmetries for Learning Program Semantics"
_ICLR.cc/2024/Conference — Submitted to ICLR 2024_

### Official Review · Reviewer_3XTL · 2023-10-26

**Soundness:** 4 excellent
**Presentation:** 3 good
**Contribution:** 4 excellent
**Rating:** 8
**Confidence:** 4

**Summary:**

The paper proposes SymC, an approach to training a Transformer architecture that is invariant/equivariant to dependence-preserving reordering of code. SymC's formal foundation is a group theoretic definition of invariance and equivariance. The paper defines this symmetry group over program interpretation graphs, graphs whose nodes are program instructions and whose edges indicate whether there is any execution path in which there is a direct dependence between data computed by the two instructions. The paper then relaxes this to the sound overapproximation of dependence graphs, and shows an implementation of self-attention that is equivariant to actions of this symmetry group. The paper evaluates the proposed model on a range of invariant tasks, evaluating on code transformations that fall within and outside the scope of the invariance. The paper shows that SymC is competitive or surpasses baseline models on nearly all metrics of interest.

**Strengths:**

* Broadly, the paper is very well written. The paper provides a clear description of all the background knowledge of symmetry groups, clearly grounds the theory in the task (symmetry groups of code ordering), explains the implementation reasonably well (invariant/equivariant self-attention), and has a clear evaluation.
* The problem domain itself is interesting and important, and the proposed solution is novel
* The evaluation is quite extensive, with strong results on all metrics of interest

**Weaknesses:**

* The one part of the paper that could be more clear is in the precise discussion of the implementation of SymC in a Transformer. Specifically:
  * I found the definition of the Aut(PDG) distance matrix to be a bit hard to reason about
  * I also wasn't sure what the relationship between this distance matrix and the standard use of positional encodings is.
* I would appreciate some discussion of why the F1/AUC results in Table 2 are not monotonic in the percentage of semantics-preserving permutations.
* The evaluation also lacks any quantification of variance in the results (e.g., standard error across different training trials).
* Minor typo: Section 3.3: "tofuture"

**Questions:**

* Does SymC use positional embeddings?
* Why are the F1/AUC results in Table 2 not monotonic in the percentage of semantics-preserving permutations?
* Could the authors provide examples of code where the relaxation of the interpretation graph to the program dependence graph is too conservative?

---

> ### Author Response · Authors · 2023-11-21
>
> We really appreciate your constructive comments. We address each of your points below.
>
> **Weakness 1.1: The definition of the Aut(PDG) distance matrix to be a bit hard to reason about**
>
> We will clarify the definition of the distance matrix used in our paper. In general, there can be various distance matrices to represent the graph in self-attention layers and preserve automorphisms as long as it satisfies the two properties of Lemma 4 & 5 (see proofs in Lemma 6 in Appendix). We chose to represent the distance between two nodes using their distance to their lowest common ancestor, based on the intuition that this can potentially represent how they are related based on program dependencies, e.g., branch condition, same source of dataflow, etc.
>
> **Q1 & Weakness 1.2: Relationship between this distance matrix and the standard use of positional encodings**
>
> SymC does not use absolute positional embeddings fed to the input sequences. Instead, it follows a similar setting as relative positional embedding, which biases the attention computation, i.e., the product of query and key, with the program structure. Unlike regular relative positional embedding, where the distance between the pair of nodes is still computed based on their relative distance in the linear sequence, e.g., the distance between token 2 and token 5 is (5-2=3), we compute the relative distance based on the customized distance matrix using PDG, and we prove the self-attention biased by this distance matrix preserves the automorphism of PDG, thus the semantics-preserving code permutations.

---

> > ### Author Response · Authors · 2023-11-21
> >
> > **Q2 & Weakness 2: unclear why the F1/AUC results in Table 2 are not monotonic in the percentage of semantics-preserving permutations**
> >
> > We do observe more statements involved in permutations do not necessarily imply more incorrect predictions. One possible explanation is that permutations involving a higher percentage of statements could represent more extensive changes in the code. Such changes might more closely resemble variations found in real-world code distributions. In contrast, single permutations, which only involve altering a single pair of statements, may be less representative of typical real-world scenarios. It is thus hard to gauge the correlation between model performance and permutation percentage.
> >
> > **Weakness 3: the evaluation also lacks any quantification of variance in the results.**
> >
> > We will update the paper by including variance. In our new results, see our responses above to Reviewer Srm2, we have included the variance by running the experiments multiple times with different random seeds.

---

> ### Author Response · Authors · 2023-11-21
>
> **Q4: Illustrate cases where the relaxation of the interpretation graph to the program dependence graph is too conservative**
>
> As described in our response to Reviewer 6HuJ, consider the case in binary analysis:
>
> ```asm
> …
> 1       mov rcx, [rbx]
> 2       mov [rax], rdx
> …
> ```
> We have 2 nodes in PDG, i.e., {1,2} denoting 2 instructions, and the edge {1->2} because we can not easily eliminate the possibility that `[rbx]` and `[rax]` would point to the same memory location. This can lead to the spurious edge, rendering the permutation of 1 and 2 an invalid automorphism. In memory-bound applications, memory-accessing instructions can be very frequent. For these programs, the size of the automorphism group can indeed be very small (not many valid semantics-preserving permutations), restricting the number of possible permutations.

---

> > ### Author Response · Authors · 2023-11-23
> >
> > Dear reviewer 3XTL,
> >
> > Thanks again for your constructive comments. We hope our responses satisfactorily addressed your concerns.

---

### Official Review · Reviewer_Srm2 · 2023-10-28

**Soundness:** 1 poor
**Presentation:** 2 fair
**Contribution:** 3 good
**Rating:** 3
**Confidence:** 5

**Summary:**

This paper introduces SymC, a novel approach that leverages code symmetries, defined through group-theoretic principles, to enhance large language models (LLMs) for program analysis. By embedding a group-equivariant self-attention mechanism within the Transformer architecture, SymC achieves significant improvements in understanding program semantics. The method demonstrates strong generalizability across various code transformations, outperforming existing state-of-the-art LLMs, including WizardCoder and GPT-4, by substantial margins in four specific tasks.

**Strengths:**

1. The paper presents a unique and innovative approach to harnessing code symmetry, grounded in group theory, which stands out from previous methods that rely on ad-hoc heuristics. Instead of using these transformations for data augmentation, as is common in prior work, SymC ingeniously incorporates them into the attention layers of Transformers, showcasing a novel application.

2. SymC's performance is noteworthy, as it surpasses the baselines across the various tasks presented in the paper, sometimes by a large margin.

**Weaknesses:**

1. The paper could benefit from a more comprehensive comparison with related works, such as DOBF (https://arxiv.org/abs/2102.07492), which exploits code symmetry in pretraining through a deobfuscation objective, and CodeT5 (https://arxiv.org/abs/2109.00859), which leverages code symmetry in pretraining with identifier-aware data augmentation. These related works were not discussed or compared to the proposed method in the paper.

2. The evaluation framework relies heavily on four artificial tasks created by the authors, omitting well-established, practical benchmarks used commonly in the field. For instance, important code generation tasks like OpenAI HumanEval  (https://huggingface.co/datasets/openai_humaneval) and MBPP (https://huggingface.co/datasets/mbpp), as well as code translation, clone detection, defect detection, and code repair tasks from CodeXGLUE (https://github.com/microsoft/CodeXGLUE), are all relevant to the domain but were not considered. This absence of evaluation on existing benchmarks and comparison with related works raises questions about the paper's soundness and the model's real-world applicability.

3. The paper does not discuss potential limitations of the SymC model, such as its requirement for input code to be processed by a parser and a static analysis tool. This requirement may limit the model's applicability when dealing with incomplete or syntactically incorrect code, such as in code completion tasks or when faced with an empty Python block. While it is acceptable to establish certain assumptions for input code, these assumptions must be explicitly discussed rather than overlooked.

**Questions:**

1. Can the authors provide a comparative analysis of SymC with related works such as DOBF and CodeT5 that also leverage code symmetry?
2. Why were the evaluation tasks limited to four artificial tasks created by the authors?

---

> ### Author Response · Authors · 2023-11-21
>
> We really appreciate your constructive comments. We address each of your points below, with additional experiments. We think that this clarification might significantly affect your view of the paper. We hope that you will take this into consideration.
>
> **Q1 & Weakness 1: The paper could benefit from a more comprehensive comparison with related works, such as DOBF (https://arxiv.org/abs/2102.07492), which exploits code symmetry in pretraining through a deobfuscation objective, and CodeT5 (https://arxiv.org/abs/2109.00859), which leverages code symmetry in pretraining with identifier-aware data augmentation. These related works were not discussed or compared to the proposed method in the paper.**
>
> Thanks for introducing the related works to us. As shown in our response to Reviewer nM8R, Q2, we include the new results comparing SymC to DOBF, CodeT5, and GraphCodeBERT. These baselines are pre-trained, while SymC is not. Therefore, to enable a more fair comparison, we consider both pre-trained and non-pre-trained baselines and also a pre-trained version of SymC (similar to the experiment setup in Section 6.3, Figure 4(b)). In particular, the pre-trained version of SymC was pre-trained using masked language modeling on the functions provided by code2seq (java-small in the paper).
>
> Not pre-trained:
>
> |               | Original Sample (F1) | Permuted Samples (F1) | Invariance Violation Rate (the lower the better) |
> |---------------|-----------------|-------------------------|----------------------|
> | SymC          | 36.3            | 36.3                    | 0%                   |
> | GraphCodeBERT | 20.83           | 20.62                   | 31%                  |
> | DOBF          | 16.34           | 20.07                   | 41%                  |
> | CodeT5        | 25.35           | 25.42                   | 1%                |
>
>
> Pre-trained:
>
> |               | Original Sample (F1) | Permuted Samples (F1) | Invariance Violation Rate (the lower the better) |
> |---------------|-----------------|-------------------------|----------------------|
> | SymC          | 37.1            | 37.1                    | 0%                   |
> | GraphCodeBERT | 28.65           | 30.27                   | 13%                  |
> | DOBF          | 34.64           | 33.71                   | 16%                  |
> | CodeT5        | 41.06           | 41.73                   | 10.7%                |
>
>
> The results in the above tables show that, without pre-training, SymC outperforms the second-best model, CodeT5, by 43.2% and 42.8%, in original and permuted test samples, respectively. The baselines have 24.3% violation rate on average when the samples are permuted, while SymC enjoys provable invariance by construction. In addition, we observe that pre-trained CodeT5 has 10.7x higher violation rate than that when it is not pre-trained. This is likely due to that the model learns spurious positional biases not robust to statement permutations during pre-training.
>
> We will add more discussion to the related work, and complete Table 2.

---

> > ### Author Response · Authors · 2023-11-21
> >
> > **Q2 & Weakness 2: The evaluation framework relies heavily on four artificial tasks created by the authors, omitting well-established, practical benchmarks used commonly in the field. For instance, important code generation tasks like OpenAI HumanEval (https://huggingface.co/datasets/openai_humaneval) and MBPP (https://huggingface.co/datasets/mbpp), as well as code translation, clone detection, defect detection, and code repair tasks from CodeXGLUE (https://github.com/microsoft/CodeXGLUE), are all relevant to the domain but were not considered. This absence of evaluation on existing benchmarks and comparison with related works raises questions about the paper's soundness and the model's real-world applicability.**
> >
> > * Regarding “artificial tasks”
> >
> > Our analysis tasks are classic program analysis tasks, and they are all adopted in practice. For example, binary code similarity detection is used for detecting vulnerabilities [1] and patching them [2], function signature prediction is used to retrofit control flow integrity [3, 4], and memory region prediction is used to facilitate malware analysis and vulnerability detection [5, 6]. We followed the baselines by using the standard datasets and experimental setup.
> >
> > [1] Pewny, Jannik, Behrad Garmany, Robert Gawlik, Christian Rossow, and Thorsten Holz. "Cross-architecture bug search in binary executables." In 2015 IEEE Symposium on Security and Privacy, pp. 709-724. IEEE, 2015.
> >
> > [2] Duan, Yue, Xuezixiang Li, Jinghan Wang, and Heng Yin. "Deepbindiff: Learning program-wide code representations for binary diffing." In Network and distributed system security symposium. 2020.
> >
> > [3] Van Der Veen, Victor, Enes Göktas, Moritz Contag, Andre Pawoloski, Xi Chen, Sanjay Rawat, Herbert Bos, Thorsten Holz, Elias Athanasopoulos, and Cristiano Giuffrida. "A tough call: Mitigating advanced code-reuse attacks at the binary level." In 2016 IEEE Symposium on Security and Privacy (SP), pp. 934-953. IEEE, 2016.
> >
> > [4] Ge, Xinyang, Nirupama Talele, Mathias Payer, and Trent Jaeger. "Fine-grained control-flow integrity for kernel software." In 2016 IEEE European Symposium on Security and Privacy (EuroS&P), pp. 179-194. IEEE, 2016.
> >
> > [5] Balakrishnan, Gogul, Thomas Reps, David Melski, and Tim Teitelbaum. "Wysinwyx: What you see is not what you execute." In Working Conference on Verified Software: Theories, Tools, and Experiments, pp. 202-213. Berlin, Heidelberg: Springer Berlin Heidelberg, 2005.
> >
> > [6] Guo, Wenbo, Dongliang Mu, Xinyu Xing, Min Du, and Dawn Song. "{DEEPVSA}: Facilitating Value-set Analysis with Deep Learning for Postmortem Program Analysis." In 28th USENIX Security Symposium (USENIX Security 19), pp. 1787-1804. 2019.

---

> > > ### Author Response · Authors · 2023-11-21
> > >
> > > * Regarding HumanEval and MBPP
> > >
> > > Our paper focuses on representing *program* semantics using code symmetry groups, and thus we focus exclusively on *program analysis tasks* that take a *program* as input. *Program synthesis tasks*, like those suggested by the reviewer (HumanEval and MBPP), often take natural language (and some code treated as also natural language prompt) as input, but we do not target symmetries in natural language in this work. Therefore, although interesting, program synthesis is out of the scope of this paper, and we defer the treatment of program synthesis through symmetries as future work.

---

> ### Author Response · Authors · 2023-11-21
>
> * New results
>
> However, we agree with the reviewer that there are many ML4code tasks, like defect prediction, that can benefit from code symmetry. To demonstrate that effect, we follow the reviewer’s suggestion by incorporating the defect prediction and include our preliminary results in the following.
>
> Particularly, we consider defects4j dataset [7] and follow CodeXGLUE to formulate the task as a binary classification (buggy/non-buggy) given the function. We ensure the number of buggy and non-buggy samples are the same, and measure the prediction accuracy on the original and permuted samples. Our baselines include GraphCodeBERT [8], DOBF [9], CodeT5 [10], UnixCoder [11], and CodeBert [12].
>
> |  | Original Sample | Permuted Sample |
> |---|---|---|
> | SymC | 68.8$\pm$1.6 | 68.8$\pm$1.6 |
> | GraphCodeBERT | 61.7$\pm$1.7 | 61.7$\pm$2.2 |
> | DOBF | 62.4$\pm$1.03 | 61.5$\pm$1.2 |
> | CodeT5 | 63.3$\pm$2.1 | 60$\pm$7.5 |
> | UnixCoder | 67.1$\pm$2.2  | 67.1$\pm$1.6 |
> | CodeBERT | 62.2$\pm$1.2 | 61.7$\pm$2.1 |
> | GraphCodeBERT-not-pre-trained | 60.6$\pm$2.4 | 61$\pm$1.7 |
> | DOBF-not-pre-trained | 59.2$\pm$1  | 59.9$\pm$1.4 |
> | CodeT5-not-pre-trained | 59.2$\pm$2.7 | 59.2$\pm$2.7 |
> | UnixCoder-not-pre-trained | 63.1$\pm$2.6 | 63.1$\pm$2.6 |
> | CodeBERT-not-pre-trained | 62.2$\pm$4.2 | 62.4$\pm$4.6 |
>
> The above table shows that SymC outperforms all the baselines, even when they are pre-trained on large-scale code datasets, by 11.4% on average. For example, SymC outperforms the second-best model, UnixCoder, by 2.5% in accuracy on permuted samples. Similar to function name prediction, all the baselines are susceptible to simple statement permutations. We will add more discussions to the related works and experiments in the paper.
>
> [7] Just, René, Darioush Jalali, and Michael D. Ernst. "Defects4J: A database of existing faults to enable controlled testing studies for Java programs." In Proceedings of the 2014 International Symposium on Software Testing and Analysis, pp. 437-440. 2014.
>
> [8] Guo, Daya, Shuo Ren, Shuai Lu, Zhangyin Feng, Duyu Tang, Shujie Liu, Long Zhou et al. "GraphCodeBERT: Pre-training code representations with data flow." arXiv preprint arXiv:2009.08366 (2020).
>
> [9] Roziere, Baptiste, Marie-Anne Lachaux, Marc Szafraniec, and Guillaume Lample. "Dobf: A deobfuscation pre-training objective for programming languages." arXiv preprint arXiv:2102.07492 (2021).
>
> [10] Wang, Yue, Weishi Wang, Shafiq Joty, and Steven CH Hoi. "Codet5: Identifier-aware unified pre-trained encoder-decoder models for code understanding and generation." arXiv preprint arXiv:2109.00859 (2021).
>
> [11] Guo, Daya, Shuai Lu, Nan Duan, Yanlin Wang, Ming Zhou, and Jian Yin. "Unixcoder: Unified cross-modal pre-training for code representation." arXiv preprint arXiv:2203.03850 (2022).
>
> [12] Feng, Zhangyin, Daya Guo, Duyu Tang, Nan Duan, Xiaocheng Feng, Ming Gong, Linjun Shou et al. "Codebert: A pre-trained model for programming and natural languages." arXiv preprint arXiv:2002.08155 (2020).

---

> ### Author Response · Authors · 2023-11-21
>
> **Weakness 3: The paper does not discuss potential limitations of the SymC model, such as its requirement for input code to be processed by a parser and a static analysis tool. This requirement may limit the model's applicability when dealing with incomplete or syntactically incorrect code, such as in code completion tasks or when faced with an empty Python block. While it is acceptable to establish certain assumptions for input code, these assumptions must be explicitly discussed rather than overlooked.**
>
> As shown in our response to Reviewer 6HuJ, statically analyzing the input code incurs additional overhead, e.g., 250.2ms per sample, but still much less than the existing model. That being said, we agree that efficiently analyzing incomplete or syntactically incorrect code on the fly can be an interesting future direction. We will add the discussion of limitations in the paper.

---

> ### Author Response · Authors · 2023-11-23
>
> Dear reviewer Srm2,
>
> Thanks so much again for your constructive comments. We would be grateful if you could let us know whether our responses, i.e., the experiments on comparing to code baselines, e.g., DOBF, CodeT5, GraphCoderBERT, CodeBERT, and UnixCoder, and the evaluation on a new task, i.e., defect prediction, satisfactorily answer your questions and if so, would you like to increase the score. Thanks!

---

### Official Review · Reviewer_nM8R · 2023-10-29

**Soundness:** 2 fair
**Presentation:** 2 fair
**Contribution:** 2 fair
**Rating:** 3
**Confidence:** 4

**Summary:**

The authors introduce a group-theoretic framework that defines code symmetries as semantics-preserving transformations, enabling precise reasoning within LLMs. SYMC employs a novel variant of group-equivariant self-attention that is provably equivariant to code symmetries.
The evaluation results show that SYMC generalizes to unseen code transformations, outperforming the state-of-the-art code models by 30.7%.

**Strengths:**

The idea of defining code symmetries as semantics-preserving transformations, enabling precise reasoning within LLMs is somewhat interesting.
To evaluate the approach, four analysis tasks that require a deep understanding of code behavior such that they are expected to stay invariant to code symmetries were considered. Also a set of real-world semantics-preserving transformations
beyond PDG automorphisms to evaluate SYMC’s generalization in the experiments.

**Weaknesses:**

The paper needs more evaluations, e.g. an evaluation of the robustness of SYMC using the adversarial attack methods based on code transformations.
Some contents are not well presented/stated.

**Questions:**

Q: As stated in the paper, current code LLMs struggle with generalization to new code. Have you tried to evaluate the robustness of SYMC using the adversarial attack methods based on code transformations? The evaluation may make your method more convincing.

Q: Have you tried to compare with "Graphcodebert: Pre-training code representations with data flow", which is a state-of-the-art method considering the inherent structure of code, in your evaluation?

Q: Page 5, "PDG (VPDG,EPDG) is a super graph of IG, sharing the same vertices but having a superset of edges (EPDG ⊇ EIG), because we consider all memory accesses as aliasing, making PDG a conservative construction of IG",
If you "consider all memory accesses as aliasing", which is apparently a very weak encoding of the program semantics, it seems there would be too many aliases in the programs, making most statements unexchangeable to accomplish semantics-preserving statement permutations?

Q: Page 6, "Each entry dij is a 2-value tuple (pij , nij), indicating the shortest path from the lowest common ancestor of Vi and Vj , denoted as Tij , to Vi and Vj , respectively", is pij the positive distances and nij the negative distances as denoted in the next paragraph? Also, what do you mean by positive distances and negative distances?

Q: Page 2, "SYMC enforces its output to stay invariant via keeping its learned representation G-equivariant, where the code representation (e1, e2, e3, e4) is transformed into (e2, e3, e4, e1) xxx", should "(e2, e3, e4, e1)" be "(e2, e1, e3, e4)" as shown in Figure 1a?

Q: Page 9, the lines labeled 2nd, 4th, 6th in Figure 4a. Which lines are for Aut(PDG)-equivariant self-attention layers and which are for the Aut(PDG)-invariant ones?

---

> ### Author Response · Authors · 2023-11-21
>
> We appreciate your time and effort in reviewing the paper. We address each of your points below.
>
> **Q1: As stated in the paper, current code LLMs struggle with generalization to new code. Have you tried to evaluate the robustness of SYMC using the adversarial attack methods based on code transformations? The evaluation may make your method more convincing.**
>
> Thanks for bringing up the robustness angle against adversarial attacks. The key approach we started off from this paper is the development of a code model with **provable robustness by construction** against semantics-preserving code permutations. Guided by this framework, we proved that our approach is verifiably robust against any possible adversarial permutations, and thus not bound by minimal perturbations assumed in many adversarial attacks [1]. We leave exploring other code symmetry groups and their robustness guarantees as future work.
>
> [1] Pierazzi, Fabio, Feargus Pendlebury, Jacopo Cortellazzi, and Lorenzo Cavallaro. "Intriguing properties of adversarial ML attacks in the problem space." In 2020 IEEE Symposium on Security and Privacy (S&P), pp. 1332-1349. IEEE, 2020.
>
> Indeed, when evaluating *other code transformations beyond permutations*, we agree that adversarial attacks can potentially lead to a greater number of violations when they are guided properly, e.g., by the input gradient. Therefore, we evaluate SymC and the newly suggested code models against the adversarial attack for function name prediction, i.e., Averloc [2]. The adversarial transformations implemented in Averloc are comprehensive, and include the adversarial code transformations proposed in prior work by Yefet et al. [3], e.g., variable renaming, dead code insertion, etc., with additional transformations such as loop unrolling. We follow the setting in Averloc by computing the adversarial attacks against a seq2seq model trained by the Averloc authors, and evaluate SymC and the baselines (GrahCodeBERT as suggested by the reviewer, and DOBF and CodeT5 as suggested by reviewer Srm2) on the generated adversarial examples. This ensures a fair comparison by evaluating all the models on the same set of adversarial examples. The baselines are trained on the same exact training set as SymC.
>
> |               | Original Sample (F1) | Adversarial Sample (F1) | Invariance Violation (the lower the better) |
> |---------------|-----------------|--------------------|----------------------|
> | SymC          | 52.9            | 47.5               | 26%                   |
> | GraphCodeBERT | 52.56           | 42.89              | 51%                  |
> | DOBF          | 51.59          | 39.68              | 51%                  |
> | CodeT5        | 44.21           | 36.66              | 47%                |
>
> The table shows that SymC outperforms the second-best baseline, GraphCodeBERT, by 10.7% and 49%, in F1 and violation rate on the adversarial examples, respectively, against adversarial transformations (beyond semantics-preserving permutations). This indicates the strong robustness of SymC against adversarial code transformations, even though the attacks are not statement permutations.
>
> [2] Henkel, Jordan, Goutham Ramakrishnan, Zi Wang, Aws Albarghouthi, Somesh Jha, and Thomas Reps. "Semantic robustness of models of source code." In 2022 IEEE International Conference on Software Analysis, Evolution and Reengineering (SANER), pp. 526-537. IEEE, 2022.
>
> [3] Yefet, Noam, Uri Alon, and Eran Yahav. "Adversarial examples for models of code." Proceedings of the ACM on Programming Languages 4, no. OOPSLA (2020): 1-30.

---

> > ### Author Response · Authors · 2023-11-21
> >
> > **Q2: Have you tried to compare with "GraphCodeBERT: Pre-training code representations with data flow", which is a state-of-the-art method considering the inherent structure of code, in your evaluation?**
> >
> > Thanks for pointing out the related works. We will add more discussion to GraphCodeBERT and other related works. To answer your specific question, we include the new evaluation results we have obtained during the rebuttal. As shown in the following, we compare SymC to the baselines including GraphCodeBERT. Significantly, SymC outperforms GrahCodeBert by 19.9%, even when it is pre-trained with substantial data augmentation effort. More importantly, SymC is provably robust to all permuted samples by construction i.e., 0% violation rate, while GraphCodeBERT (even pre-trained) has 13% of the labels changed under semantics-preserving permutations.
> >
> > | | Original Sample (F1) | Permuted Samples (F1) | Invariance Violation Rate (the lower the better) |
> > | ----------|----------|----------|----------|
> > | SymC | 36.3 | 36.3 | 0 |
> > | GraphCodeBERT | 28.65 | 30.27 | 13 |
> > | GraphCodeBERT-no-pretrain | 20.83 | 20.62 | 31 |
> >
> > **Q3: Page 5, "PDG ($V_{PDG}$,$E_{PDG}$) is a super graph of $\mathcal{IG}$, sharing the same vertices but having a superset of edges ($E_{PDG}\subseteq E_{\mathcal{IG}}$), because we consider all memory accesses as aliasing, making PDG a conservative construction of $\mathcal{IG}$", If you "consider all memory accesses as aliasing", which is apparently a very weak encoding of the program semantics, it seems there would be too many aliases in the programs, making most statements unexchangeable to accomplish semantics-preserving statement permutations?**
> >
> > We will clarify this point by adding concrete numbers showing the percentage of memory access instructions in the program. This could also depend on the purposes of the program, e.g., computation-heavy programs employ arithmetic operations and leverage the CPU registers more. We observe that highly optimized binaries often have fewer memory accesses due to the compiler optimizations that aim to make the most efficient use of CPU registers.

---

> > > ### Author Response · Authors · 2023-11-21
> > >
> > > **Q3: Page 5, "PDG ($V_{PDG}$,$E_{PDG}$) is a super graph of $\mathcal{IG}$, sharing the same vertices but having a superset of edges ($E_{PDG}\subseteq E_{\mathcal{IG}}$), because we consider all memory accesses as aliasing, making PDG a conservative construction of $\mathcal{IG}$", If you "consider all memory accesses as aliasing", which is apparently a very weak encoding of the program semantics, it seems there would be too many aliases in the programs, making most statements unexchangeable to accomplish semantics-preserving statement permutations?**
> > >
> > > This is a great point. We will include additional investigation into this problem by adding concrete numbers showing the percentage of memory access instructions in the program. We agree there would be a tradeoff between efficiency and precision, but we employ the conservative strategy in this paper for simplicity and efficiency. This could also depend on the purposes of the program, e.g., computation-heavy programs employ arithmetic operations and leverage the CPU registers more. We observe that highly optimized binaries often have fewer memory accesses due to the compiler optimizations that aim to make the most efficient use of CPU registers.
> > >
> > > **Q4: Page 6, "Each entry $d_{ij}$ is a 2-value tuple ($p_{ij}$, $n_{ij}$), indicating the shortest path from the lowest common ancestor of $V_i$ and $V_j$ , denoted as $T_{ij}$ , to $V_i$ and $V_j$ , respectively", is $p_{ij}$ the positive distances and $n_{ij}$ the negative distances as denoted in the next paragraph? Also, what do you mean by positive distances and negative distances?**
> > >
> > > Sorry for the confusion. Yes, the positive and negative distance corresponds to $p_{ij}$ and $n_{ij}$, respectively. The positive distance denotes the distance from node $V_i$ to the lowest common ancestor of node i and j, $T_{ij}$. The negative distance denotes the distance from node $V_j$ to $T_{ij}$. We will make the explanation more clear in the paper.

---

> > > > ### Author Response · Authors · 2023-11-21
> > > >
> > > > **Q5: Page 2, "SYMC enforces its output to stay invariant via keeping its learned representation G-equivariant, where the code representation $(e_1, e_2, e_3, e_4)$ is transformed into $(e_2, e_3, e_4, e_1)$ xxx", should "$(e_2, e_3, e_4, e_1)$" be "$(e_2, e_1, e_3, e_4)$" as shown in Figure 1a?**
> > > >
> > > > Great catch! We have fixed the mistake in the updated paper.
> > > >
> > > > **Q6: Page 9, the lines labeled 2nd, 4th, 6th in Figure 4a. Which lines are for Aut(PDG)-equivariant self-attention layers and which are for the Aut(PDG)-invariant ones?**
> > > >
> > > > Figure 4a shows the performance drop when we set the subsequent self-attention layers Aut(PDG)-*invariant*, not Aut(PDG)-*equivariant*, starting from 2nd, 4th, 6th self-attention layer (see caption of Figure 4). This is to demonstrate having equivariance in the code representation learning is important, compared to making it invariant throughout all the layers. We will clarify this in the paper.

---

> ### Author Response · Authors · 2023-11-23
>
> Dear reviewer nM8R,
>
> Thanks again for making the effort to review our paper. We would be grateful if you could let us know whether our responses, i.e., the experiments on adversarial attacks and GraphCodeBERT, satisfactorily answer your questions and if so, would you like to increase the score. Thanks!

---

### Official Review · Reviewer_6HuJ · 2023-11-02

**Soundness:** 3 good
**Presentation:** 2 fair
**Contribution:** 3 good
**Rating:** 6
**Confidence:** 2

**Summary:**

This work explores invariance to symmetries in code that do not change the semantics of the code. This notion is formalized via automorphisms of program interpretation graphs. To achieve equivariance (and invariance) to these automorphisms, the authors use a self-attention based model with pairwise features based on an invariant distance matrix.

**Strengths:**

1. Formalization of code symmetries as automorphisms of graphs is nice and seems like the correct formalism.
2. SymC model achieves equivariance to the code symmetries under consideration in a natural way, which is not too different from existing Transformer-based models.
3. Empirical results show that SymC outperforms strong baselines, while being small and robust to code symmetries.

**Weaknesses:**

1. Hard to understand exactly what program interpretation graphs and program dependence graphs look like, which is crucial to the paper.
2. Experimental details are lacking. What is the training procedure for SymC, is it just supervised training on the downstream task? How about for the other models? For Function Name prediction, do the LLMs take in just the text as input, and what exactly does SymC take as input there?
3. Computation of graphs associated to code may be costly and restrictive.

**Questions:**

1. Could you illustrate example program interpretation graph and program dependence graphs? This would be quite helpful for understanding.
2. How costly is obtaining the code graphs?
3. Why does SymC require "40,162 lines of code"? I'm curious as to what makes it require so much.

---

> ### Author Response · Authors · 2023-11-21
>
> We really appreciate your constructive comments. We address each of your points below.
>
> **Q1 & Weakness 1: Could you illustrate an example program interpretation graph and program dependence graphs? This would be quite helpful for understanding.**
>
> Consider the following Java example:
>
> ```java
> ……
> 1       int a = 1;
> 2       int b = 2;
> 3       if (a > b) {
> 4           return 1;
> 5       } else {
> 6           return 0;
> 7       }
> ……
> ```
> We will have 5 nodes in IG, i.e., {1,2,3,4,6} denoting 5 statements, and the edges {1->3, 2->3, 3->4, 3->6}. In such a case, IG is equivalent to PDG.
>
> Consider the case in binary analysis:
>
> ```asm
> …
> 1       mov rcx, [rbx]
> 2       mov [rax], rdx
> …
> ```
> We have 2 nodes in IG, i.e., {1,2} denoting 2 instructions, and there is no edge in actual IG, as there is no dataflow from 1 to 2. However, in PDG, we conservatively consider the edge {1->2} by marking them as a write-after-read dependence. This is because we can not easily eliminate the possibility that `[rbx]` and `[rax]` would point to the same memory location.
>
> **Weakness 2: Experimental details are lacking. What is the training procedure for SymC, is it just supervised training on the downstream task? How about the other models? For Function Name prediction, do the LLMs take in just the text as input, and what exactly does SymC take as input there?**
>
> SymC adopts only supervised training and trains the model from scratch *without any pre-training*. While Figure 4(b) demonstrates that adding pre-training brings extra improvement, we aim to demonstrate having symmetry-preserving architecture alone leads to better performance than many pre-trained models.
>
> For other baselines, we respect their own training paradigm. In particular, PalmTree, GPT4, and WizardCoder are pre-trained on programs in the wild. code2vec, code2seq, and GGNN use supervised training. The new models that we have experimented with during the rebuttal, i.e., CodeBert, GraphCodeBERT, UnixCoder, DOBF, and CodeT5 (see our responses to reviewer Srm2), are pre-trained with large-scale code datasets.
>
> For function name prediction, both LLMs and SymC take as input the function (function body and function signature) with the function name stripped.

---

> ### Author Response · Authors · 2023-11-21
>
> **Q2 & Weakness 3: How costly is obtaining the code graphs? Computation of graphs associated to code may be costly and restrictive.**
>
> To incorporate the inductive bias of code, many code models involve extracting and encoding code structures, including our baselines, as computing PDGs statically is not overly expensive. Below, we add the runtime performance (in milliseconds) per code sample of SymC and PalmTree on binary analysis, training, and inference, using the same exact hardware (an AMD EPYC 7502 processor, 128 virtual cores, 256GB RAM, Nvidia RTX 3090 GPUs).
>
> | | Graph Construction | Train | Inference |
> | ----------|----------|----------|----------|
> | SymC | 250.2ms | 15.4ms | 2ms |
> | PalmTree | 4460ms | 36ms | 17ms |
>
> SymC’s cheap computation of PDG incurs 17.8x less runtime overhead. It is also an interesting research problem to incorporate optimization, e.g., caching, to improve the efficiency of the PDG computation during inference. We will update the paper to include both source and binary analysis tasks.
>
>
> **Q3: Why does SymC require "40,162 lines of code"? I'm curious as to what makes it require so much.**
>
> Besides implementing the model and algorithm (~4.5k lines), we include the lines of code for doing program analysis (both source and binary code) for obtaining PDGs (2.4k lines) and generating ground truths for the analysis tasks (3k lines). We also include the code for setting up the baselines (4.3k lines), implementing program transformations (2.8k lines), data processing (1k), and all the scripts for experiments and result analysis (22k lines). We will clarify in the paper by providing a breakdown for each of these components.

---

> > ### Comment · Reviewer_6HuJ · 2023-11-22
> >
> > Dear authors,
> >
> > Thanks for the comments. The example and runtimes are helpful, I definitely recommend putting them in the new versions of the paper.
> >
> > One issue I have with the paper is the framing of SymC as an LLM (this is why I asked the question about the training procedure). The model is a Transformer with graph information included via relative positional encodings. But it cannot be trained or used for next-token prediction, but it can be used for supervised tasks on parts of code.
> >
> > Relatedly, models like GGNN may be a better comparison. Notably that model has lower invariance violation than LLMs. Given that GGNN also works on graphs associated to code, could you explain why it is not invariant to the code symmetry transformations you consider?

---

> ### Author Response · Authors · 2023-11-22
>
> **Q1: The example and runtimes are helpful, I definitely recommend putting them in the new versions of the paper.**
>
> We are grateful for your prompt response. We will definitely include them in the paper - we are working on drafting the figure/table and adding the discussion. We will update you as soon as we finish the draft.
>
> **Q2: One issue I have with the paper is the framing of SymC as an LLM (this is why I asked the question about the training procedure). The model is a Transformer with graph information included via relative positional encodings. But it cannot be trained or used for next-token prediction, but it can be used for supervised tasks on parts of code.**
>
> We appreciate you pointing out this issue. We agree that our focus in this work is primarily on the Transformer encoder with bidirectional self-attention, without explicitly demonstrating its application in the context of a Transformer decoder, as commonly used in most LLMs today.
>
> However, we would like to highlight Section 3.5, “Token-level predictor” and Lemma 3, and the proof in Appendix B in the paper. We demonstrate how SymC preserves $Aut(\mathcal{IG})$-invariance at the individual token level. This ensures end-to-end $Aut(\mathcal{IG})$-invariant property in a Transformer decoder. As the autoregressive decoder generates a single token one at a time, attending to all previous tokens is in the same manner as the token-level predictor, and it is $Aut(\mathcal{IG})$-invariant according to Lemma 3.
>
> We will edit the framing of the paper, by focusing more on how SymC augments the Transformer architecture, and leave a detailed evaluation of LLM (autoregressive decoder) as future work. We will edit Section 3.5 to make the discussion of Transformer decoder more explicit. Thanks for helping us improve the clarity of our paper.

---

> ### Author Response · Authors · 2023-11-22
>
> **Q3: Relatedly, models like GGNN may be a better comparison. Notably that model has lower invariance violation than LLMs. Given that GGNN also works on graphs associated to code, could you explain why it is not invariant to the code symmetry transformations you consider?**
>
> We will answer this question from both conceptual and experimental perspectives.
>
> **Conceptually**, while GGNN incorporates the data and control flow graph, its particular dataflow construction *does not preserve* invariance to semantics-preserving permutations. For example, GGNN maintains edges for all reads and writes to the same variable token, e.g., using LastRead and LastWrite edge types [1]. This will prevent valid permutation between two statements that only read the same variable (Read-after-Read). Moreover, the permuted nodes can potentially lead to changed edge types. Such designs can result in the changed model’s output while its input transformation is semantics-preserving, breaking the robustness guarantees that SymC offers.
>
> [1] Allamanis, Miltiadis, Marc Brockschmidt, and Mahmoud Khademi. "Learning to represent programs with graphs." arXiv preprint arXiv:1711.00740 (2017).
>
> However, we agree that GNNs based strictly on our PDG will naturally preserve the Aut(PDG)-invariance. We aim to study how such a GNN architecture compares to SymC in our future work. The line of works on the graph Transformer suggests that Transformer with graph structures outperforms both the vanilla Transformer and GNN [2,3], but it remains interesting to study their robustness guarantees to code [3].
>
> [2] Ying, Chengxuan, Tianle Cai, Shengjie Luo, Shuxin Zheng, Guolin Ke, Di He, Yanming Shen, and Tie-Yan Liu. "Do transformers really perform badly for graph representation?." Advances in Neural Information Processing Systems 34 (2021): 28877-28888.
>
> [3] Hellendoorn, Vincent J., Charles Sutton, Rishabh Singh, Petros Maniatis, and David Bieber. "Global relational models of source code." In International conference on learning representations. 2019.
>
> **Experimentally**, we observe the particular low-invariance-violation on GGNN in our experiments is because it keeps predicting the same meaningless label, i.e., its F1 score remains at 0.016 across all the permutation percentages (Table 2). For example, in more than 60% of the samples, GGNN implementation provided by [4] (the original GGNN paper [1] did not perform function name prediction) keeps predicting a meaningless name “SPR9486” on disparate samples.
>
> [4] Rabin, Md Rafiqul Islam, Nghi DQ Bui, Ke Wang, Yijun Yu, Lingxiao Jiang, and Mohammad Amin Alipour. "On the generalizability of Neural Program Models with respect to semantic-preserving program transformations." Information and Software Technology 135 (2021): 106552.

---

> > ### Author Response · Authors · 2023-11-22
> >
> > Moreover, even GGNN keeps its output invariant by predicting the same incorrect label most of the time, it still changes its prediction under simple semantics-preserving permutations. Here is one example in the test set, where GGNN predicts “*run*”:
> >
> > ```Java
> > 1  private void applyNormalPattern(String modeStr, Matcher matcher) {
> > 2      boolean commaSeperated = false;
> > 3      for (int i = 0; i < 1 || matcher.end() < modeStr.length(); i++) {
> > 4          if (i > 0 && (!commaSeperated || !matcher.find())) {
> > 5              throw new IllegalArgumentException(modeStr);
> > 6          }
> > 7          String str = matcher.group(2);
> > 8          char type = str.charAt(str.length() - 1);
> > 9          boolean user, group, others, stickyBit;
> > 10        user = group = others = stickyBit = false;
> > 11  }
> > ```
> >
> > By permuting line 9 to the line before lines 7 and 8, which does not change the behavior of the code at all:
> >
> > ```Java
> > 1  private void applyNormalPattern(String modeStr, Matcher matcher) {
> > 2      boolean commaSeperated = false;
> > 3      for (int i = 0; i < 1 || matcher.end() < modeStr.length(); i++) {
> > 4          if (i > 0 && (!commaSeperated || !matcher.find())) {
> > 5              throw new IllegalArgumentException(modeStr);
> > 6          }
> > 7          boolean user, group, others, stickyBit;
> > 8          String str = matcher.group(2);
> > 9          char type = str.charAt(str.length() - 1);
> > 10        user = group = others = stickyBit = false;
> > 11  }
> > ```
> >
> > GGNN predicts “*update*”.
> >
> > We will add more discussions on graph-based approaches for code modeling in the paper.

---

### Meta-Review · Area_Chair_pS43 · 2023-12-06

**Metareview:**

This paper introduces SymC, a novel approach that leverages code symmetries, defined through group-theoretic principles, to enhance large language models (LLMs) for program analysis. The paper defines a symmetry group over program interpretation graphs, and then relaxes this to the sound overapproximation of dependence graphs. SymC achieves significant improvements by embedding a group-equivariant self-attention mechanism within the Transformer architecture. The empirical results show that SymC is competitive or surpasses baseline models, including WizardCoder and GPT-4, in four specific tasks. Most of the reviewers would have appreciated more details about the program graphs and their construction. An insufficient experimental comparison with previous work was also mentioned in multiple reviews.

**Justification For Why Not Higher Score:**

The paper introduces an interesting idea but there seems to be substantial gaps in the exposition. There was some confusion among the reviewers about what exactly SymC is, and if the framework was scalable. More experimental comparisons to previous work has also been requested.

**Justification For Why Not Lower Score:**

N/A

---

### Decision · Program_Chairs · 2024-01-16

Reject